# Entropy and Wealth

**DOI:** 10.3390/e23101356

**Published:** 2021-10-17

**Authors:** Demetris Koutsoyiannis, G.-Fivos Sargentis

**Affiliations:** Department of Water Resources and Environmental Engineering, School of Civil Engineering, National Technical University of Athens, 15780 Athens, Greece; fivos@itia.ntua.gr

**Keywords:** entropy, wealth, income distribution, options, potentiality, principle of maximum entropy, Second Law, stochastics

## Abstract

While entropy was introduced in the second half of the 19th century in the international vocabulary as a scientific term, in the 20th century it became common in colloquial use. Popular imagination has loaded “entropy” with almost every negative quality in the universe, in life and in society, with a dominant meaning of disorder and disorganization. Exploring the history of the term and many different approaches to it, we show that entropy has a universal stochastic definition, which is not disorder. Hence, we contend that entropy should be used as a mathematical (stochastic) concept as rigorously as possible, free of metaphoric meanings. The accompanying principle of maximum entropy, which lies behind the Second Law, gives explanatory and inferential power to the concept, and promotes entropy as the mother of creativity and evolution. As the social sciences are often contaminated by subjectivity and ideological influences, we try to explore whether maximum entropy, applied to the distribution of a wealth-related variable, namely annual income, can give an objective description. Using publicly available income data, we show that income distribution is consistent with the principle of maximum entropy. The increase in entropy is associated to increases in society’s wealth, yet a standardized form of entropy can be used to quantify inequality. Historically, technology has played a major role in the development of and increase in the entropy of income. Such findings are contrary to the theory of ecological economics and other theories that use the term entropy in a Malthusian perspective.


**Dedicated to the memory of Themistocles Xanthopoulos**

whose book series “Requiem with Crescendo?” [1,2,3] triggered [4] this paper

*Wealth is not about having a lot of money; it’s about having a lot of options*
Chris Rock (American comedian, writer, producer and director)

*Πάντα τὰ ἐμὰ μετ’ ἐμοῦ φέρω (All that is mine I carry with me)*
Bias of Priene (one of the seven Greek sages; 6th c. BC;quoted in Latin by Cicero, Paradoxa Stoicorum I, 8, as “Omnia mea mecum porto”)

## 1. Introduction

The word “entropy” was introduced about 150 years ago as a scientific term, but later its use became common in everyday language. We can find it in literature [5,6], in poetry [7,8], in the press, and in web posts, but often its use is irrelevant to its real scientific meaning. The most common use of the word entropy is when a writer wants to describe, with an “intellectual” word, a kind of disorder. We will clarify in detail in Section 2.3 that this is a misinterpretation of the actual meaning of the term, which in fact is more related to uncertainty. The wide colloquial use of entropy becomes clear if we consider the detailed information available in the over 60,000 words (lemmas) of English, based on data from the Corpus of Contemporary American English (COCA), whose content includes eight genres: spoken, fiction, popular magazines, newspapers, academic texts and more [9]. According to data given by Word and Phrase Info [10] and plotted in Figure 1, the frequency of “entropy” being used in fiction, for example, is not dramatically lower than its use in academic texts.

The academic corpus can be investigated using bibliometric databases, such as Scopus [11]. The results obtained from several searches in the latter are shown in Figure 2. One would expect that the term “entropy” would more frequently appear in scholarly articles in combination with terms such as “physics” or “thermodynamics”—and indeed this is the case for recent years. Amazingly, however, before the 1960s, the combination of the term “entropy” with “society” or “social” was more frequent than the former. This suggests the appeal of a concept related to the Second Law of thermodynamics in social sciences. We note that in that period, the probabilistic content of entropy (see Section 2) was not fully developed, and thus, in papers before the 1960s, entropy was used with its classical thermodynamic meaning (also explained in Section 2). Specifically, the resource flow in economics was parallelized with the energy flow in thermodynamics. As seen in the figure, in the 21st century, “entropy” is also used in combination with ecology and economics.

Out of its physical and stochastic context, the term “entropy” is typically used metaphorically, and hence its meaning becomes ambiguous or diverse. For example, the term “social entropy”, in one of its earliest uses in scholarly publications [12], is equated to “dereliction, pollution and waste” created by “economic activity” or by “society as consumers”, which have to be minimized. Bailey, in his book entitled “Social Entropy Theory” [13], tried to illuminate
*the fundamental problems of societal analysis with a nonequilibrium approach, a new frame of reference built upon contemporary macrological principles, including general systems theory and information theory.*
His interest was more in illuminating “Social Entropy Theory” than in defining social entropy per se. Nor in an overview of his book [14] did Bailey provide a definition of social entropy. In a critique of the book, Mayer [15] found the “unrelenting abstractness of Social Entropy Theory quite frustrating”, and added:


*Never is the theory applied to real sociological data or anything like a real social situation.*


Neto et al. [16], who also used Bailey’ theory, provided little help in clarifying what social entropy is. Recently, Dinga et al. [17], building on Bailey’s theory, clarified, rather qualitatively, the concept of social entropy as follows: (i) it cannot be of a thermodynamic type; (ii) it must be connected with social order; (iii) its connection with social order must be inversely proportional; (iv) it must hold connotations of the relationship between the homogeneity and heterogeneity of a system/process; and (v) it is fundamentally grounded by normativity. Interestingly, however, Balch [18] used the term “social entropy” as a measure of robot group diversity, proposing a formal mathematical definition.

Different aspects of entropy and energy in social systems have been examined by Mavrofides et al. [19], as well as by Davis [20], who examined how the physical and mathematical notions of entropy can be usefully imported into the social sphere. Davis also used the term “social entropy” without a definition. Further, he stated:
*Entropy has been characterized variously and vaguely by the words decadence, decay, anarchy, dystopia, randomness, chaos, decay, waste, inefficiency, dissipation, loss of available energy, irreversibility, indeterminacy, most probable states, equilibrium, thermal death* […]*. In the social sphere it has been characterized as apocalypse, disorder, disorganization, disappearance of distinctions, meaninglessness, absurdity, uncertainty, pandemonium, loss of information, inert uniformity, incoherence.* […] *In* [humanistic] *areas, the concept is used more or less as a metaphor or a synonym for chaos, disorder, breakdowns, dysfunctions, waste of material and energy, enervation, friction, inefficiencies.*
Davis was influenced by Saridis [21], according to whom “entropy measures the waste produced when work is done for the improvement of the quality of human life”; he highlighted the following quotation by Saridis:


*The concept of Entropy creates a pessimistic view for the future of our universe. The equalization of all kinds of sources of activities is leading to the equivalent of thermal death and universal boredom of our world.*


In economics, Frederick Soddy (1877–1956) [22,23,24] and Nicholas Georgescu-Roegen (1906–1994) [25,26,27], fascinated by entropy in thermodynamics, sought analogies with economics and development from a Malthusian perspective. Avery [28] noted that:


*Early in the 20th century, both Frederick Soddy and Nicholas Georgescu-Roegen discussed the relationship between entropy and economics. Soddy called for an index system to regulate the money supply and a reform of the fractional reserve banking system, while Georgescu-Roegen pointed to the need for Ecological Economics, a steady-state economy, and population stabilization.*


Following these, a series of papers and books studied similarities between economics and thermodynamic entropy [25,29,30,31,32,33,34,35,36,37,38,39,40,41,42]. McMahon and Mrozek used entropy, within the context of neoclassical economic thought, as a limit to economic growth [43]. In the same spirit, Smith and Smith used the Second Law of thermodynamics to again determine limits to growth [44]. In a review paper, Hammond and Winnett [45] presented the influence of thermodynamics on the emerging transdisciplinary field of ecological economics. However, Kovalev [46] claimed that entropy cannot be used as a measure of economic scarcity.

In a different context, in their editorial note in a special issue on “Maximum Entropy Economics: Foundations and Applications”, Scharfenaker and Yang [47] asked: “Maximum entropy economics: where do we stand?” In reply, the same authors [48] offered a brief overview of what they considered the state of maximum entropy reasoning in economic research. This involved a probabilistic (or information-based) definition of entropy. Likewise, Ryu [49], presented a technique to determine the functional forms of income distributions maximizing entropy under given conditions. Fu et al. [50] used entropy divergence methods to define measures of income inequality; notably, they regard the uniform distribution of income as the one with “the least inequality” (see discussion on this in Section 4.1 below). Mayer et al. [51] provided a theoretical framework for studying the dynamics of coupled human and natural systems in an attempt to define sustainability.

Related to these developments is the interdisciplinary research field of econophysics [52] (apparently synthesized from the Greek words “οικονομία/economy” and “φυσική/physics”, albeit violating the rules of word synthesis, as the correct use would result in “economophysics”; note that the omitted syllable “mo” distorts the intended meaning because it is “νόμο(ς)/nomo(s)” the part of the word that means “law”, while “νο/no” means nothing). In their comprehensive book entitled “Econophysics of income and wealth distributions”, Chakrabarti et al. [53] discuss entropy maximization in several forms, inspired by physics and applied to economics. Additional reviews of the field can be found in [48,54,55,56,57,58] and references therein.

Apparently, the above overview of entropy in social sciences and particularly in economics is not complete. Our purpose is not to review all related works, but to highlight two facts. First, that the use of the notion of entropy is mainly metaphorical, rich in imaginary interpretation, and divergent. Second, that the dominant view is that entropy epitomizes all “bad things” one can think of in the universe, in life, in human societies and in economics.

Our own view is quite different. On the first issue, we insist that entropy should be used as a mathematical (in particular, stochastic) concept as rigorously as possible. We avoid using ambiguous terms such as “social entropy”. We claim that any interpretation of entropy should be as close to the mathematical definition as possible, and free of metaphoric meanings. On the second issue, we believe that the overloading of the concept of entropy with negative properties reflects a misunderstanding of the underlying theory, guided by a deterministic world view—in which, however, entropy has no place.

We clarify our own view of entropy and its meaning in Section 2, after tracing its roots as a scientific concept and its historical evolution in the last 150 years. In Section 3, we provide a formal presentation of the principle of maximum entropy and its results under conditions relevant to material wealth. In Section 4 we apply the framework to the economy, trying to show that the principle of maximum entropy explains the general behaviors seen in economics.

We try to provide several theoretical and even philosophical insights on important issues related to entropy and economy, which are unavoidably influenced by our own perception—clearly an optimistic one, contrary to the pessimism expressed in most of the papers reviewed above. At the same time, we try to form insights based on real-world data, rather than speculation. The data we use are freely available on the internet and the reader can retrieve them and reproduce our calculations, or check and reprocess them independently.

Finally, we try to make the paper self-contained and independent, so that even a reader unfamiliar with entropy, with only a basic knowledge of calculus and probability, could understand it. The mathematical framework we develop can readily be put to work on the simplest computational framework (e.g., a spreadsheet).

## 2. What Is Entropy?

### 2.1. The Origin of the Entropy Concept

The word *ἐντροπία* (Greek for entropy) appears in ancient Greek [59] (from the verb ἐντρέπειν, “to turn into, to turn about”) but was only introduced into the international scientific vocabulary by Rudolf Clausius in 1865 (although the concept also appears in his earlier works, as described in [60]). The rationale for introducing the term is explained in his own words [61] (p. 358), which indicate that he was not aware of the existence of the word in ancient Greek:

*We might call S the transformational content of the body *[…]*. But as I hold it to be better to borrow terms for important magnitudes from the ancient languages, so that they may be adopted unchanged in all modern languages, I propose to call the magnitude S the entropy of the body, from the Greek word τροπή, transformation. I have intentionally formed the word entropy so as to be as similar as possible to the word energy; for the two magnitudes to be denoted by these words are so nearly allied their physical meanings, that a certain similarity in designation appears to be desirable.*

In addition to its semantic content, this quotation contains a very important insight: the recognition that entropy is related to transformation and change, and the contrast between entropy and energy, whereby the latter is a quantity that is conserved in all changes. This meaning has been more clearly expressed in Clausius’ famous aphorism [62]:


*Die Energie der Welt ist constant. Die Entropie der Welt strebt einem Maximum zu.*



*(The energy of the world is constant. The entropy of the world strives to a maximum).*


In other words, entropy and its ability to increase (as contrasted to energy, momentum and other quantities that are conserved) is the driving force of change. This property of entropy is acknowledged only rarely [63,64,65]. Instead, as we have already seen, in common perception entropy epitomizes all “bad things”, as if it were disconnected from change, or as if change can only have negative consequences, always leading to deterioration.

Mathematically, thermodynamic entropy, *S*, is defined in the same texts by Clausius through the equation dS= δQ/T, where Q and T denote heat and temperature. The definition, however, applies to a reversible process only. In an irreversible process, dS> δQ/T, which makes the definition imperfect and affected by circular reasoning, as, in turn, a reversible process is one in which the equation holds.

Two decades later (in 1877), Ludwig Boltzmann [66] (see also Swendsen [67]) gave entropy a probabilistic content as he linked it to probabilities of statistical mechanical system states, thus explaining the Second Law of thermodynamics as the tendency of the system to run toward more probable states, which have higher entropy. The probabilistic concept of entropy was advanced later in thermodynamics by Gibbs [68].

The next important step was made by Shannon in 1948 [69]. Shannon used an essentially similar, albeit more general, definition describing information content, which he also called entropy, at von Neumann’s suggestion [70,71,72]. According to the latter definition, entropy is a probabilistic concept, a measure of information or, equivalently, uncertainty. In the same year, Wiener, in his famous book *Cybernetics*, also published in 1948 [73], used the same definition for information, albeit with a negative sign (p. 62) because he regarded information as the negative of entropy (p. 11). (Interestingly, he formed the celebrated term *Cybernetics* from the Greek word *κυβερνήτης*, meaning steersman, pilot, skipper, or governor, albeit incorrectly spelling it in his book—p. 11—as *χυβερνήτης*.)

A few years later, in 1956, von Neumann [74] obtained virtually the same definition of entropy as Shannon, in a slightly different manner. Notably, as von Neumann, in addition to being a mathematician and computer scientist, was also a physicist, engineer and polymath, he clearly understood the connection of the probabilistic definition of entropy with its pre-existing physical content. Specifically, he wrote:
*An important observation about this definition is that it bears close resemblance to the statistical definition of the entropy of a thermodynamical system.* […] *Pursuing this, one can construct a mathematical theory of the communication of information patterned after statistical mechanics.*
He also cited an earlier work (1929) in physics by Szilard [75], who implied the same definition of entropy in a thermodynamic system.

The last fundamental contribution to the entropy concept was made a year later (in 1957) by Jaynes [76], who introduced the *principle of maximum entropy*. This postulates that the entropy of a stochastic system should be at maximum, under some conditions, formulated as constraints, which incorporate the information that is given about this system. This principle can be used for logical inference as well as for modeling physical systems. In this respect, the tendency of entropy to become maximal (Second Law of thermodynamics), which drives natural change, can result from this principle. On the other hand, the principle equips the entropy concept with a powerful tool for logical inference.

### 2.2. Are Thermodynamic and Probabilistic Entropy Different?

More than 150 years after the introduction of the entropy concept, its meaning is still debated, and a diversity of opinions among experts is still encountered [77]. In particular, despite having the same name, probabilistic (or information) entropy and thermodynamic entropy are still regarded by many (perhaps the majority of scientists) as two distinct notions, having in common only the name. The classical definition of thermodynamic entropy (as above) does not give any hint of its similarity with probabilistic entropy. The fact that the latter is a dimensionless quantity and the former has units (J/K) has been regarded as an argument that the two are dissimilar. Even Jaynes (2003), the founder of the maximum entropy principle, stated:


*They should never have been called by the same name; the experimental entropy makes no reference to any probability distribution, and the information entropy makes no reference to thermodynamics. Many textbooks and research papers are flawed fatally by the author’s failure to distinguish between these entirely different things.*


However, the units of thermodynamic entropy are only an historical accident, related to the arbitrary introduction of temperature scales [65]. Furthermore, the connection of probabilistic and thermodynamic entropy is clearly implied by its pioneers, Boltzmann [66], Gibbs [68], Szilard [75] and von Neumann [74]. More recent accounts of the connection have been provided by Robertson [70] and Moore [78]. Furthermore, as has recently been shown [79,80], the thermodynamic entropy of gases can be easily inferred from formal probability theory, without the need for strange assumptions (e.g., indistinguishability of particles). Impressive examples of deductive reasoning used for deriving thermodynamic laws from the formal probabilistic principle of maximum entropy have been provided in [80]. Notable among them is the derivation of the law of the phase transition of water (Clausius–Clapeyron equation) by maximizing entropy, i.e., uncertainty, at the microscopic level of a single water molecule, leading to an expression that is virtually certain at the macroscopic level.

### 2.3. Does Entropy Measure Disorder?

As already mentioned, in the public perception entropy, has a negative content, and is typically identified with disorganization or disorder and deterioration. This misleading perception has its roots in the scientific community, albeit not with the founders of the concept (except one, as we will see). Boltzmann did not identify entropy with disorder, even though he used the latter word in a footnote appearing in two papers of his [81,82], in which he speaks about the


*agreement of the concept of entropy with the mathematical expression of the probability or disorder of a motion.*


Clearly, he speaks about the irregular motion of molecules in the kinetic theory of gases, for which his expression makes perfect sense. Boltzmann also used the notion of disorder with the same meaning, in his Lectures on Gas Theory [83]. On the other hand, Gibbs [68], Shannon [69] and von Neumann [74] did not use the terms disorder or disorganization at all.

One of the earliest (in 1944) uses of the term disorder is in a paper by Darrow [84], in which he states:


*The purpose of this article has been to establish a connection between the subtle and difficult notion of entropy and the more familiar concept of disorder. Entropy is a measure of disorder, or more succinctly yet, entropy is disorder: that is what a physicist would like to say.*


Epistemologically, it is interesting that a physicist prefers the “more familiar” but fuzzy concept of disorder over the “subtle and difficult”, yet well-defined at his time, concept of entropy.

However, it appears that Wiener was the most influential scientist to support the disorder interpretation. In 1946, he gave a keynote speech at the New York Academy of Sciences [85], in which he declared that:
*Information measures order and entropy measures disorder.*
Additionally, in his influential book *Cybernetics* [73] (p. 11), he stated that
*the entropy of a system is a measure of its degree of disorganization*
wherein he replaced the term “disorder” with “disorganization”, as in this book he extensively used the former term for mental illness.

Even in the 21st century, the disorder interpretation is dominant. For example, Chaitin [86] stated:
*Entropy measures the degree of disorder, chaos, randomness, in a physical system. A crystal has low entropy, and a gas (say, at room temperature) has high entropy.*
More recently, Bailey [87] claimed:
*As a preliminary definition, entropy can be described as the degree of disorder or uncertainty in a system. If the degree of disorder is too great (entropy is high), then the system lacks sustainability. If entropy is low, sustainability is easier. If entropy is increasing, future sustainability is threatened.*
It is relevant to remark that in the latter quotations disorder has been used as equivalent to uncertainty or randomness—where the latter two terms are in essence identical [88]. Furthermore, the claim that a high-entropy system lacks sustainability is at least puzzling, given that the highest entropy occurs when a system is in the most probable (and hence most stable) state (cf. [78]).

Interestingly, Atkins [64] also explained entropy as disorder. Additionally, he noted:


*That the world is getting worse, that it is sinking purposelessly into corruption, the corruption of the quality of energy, is the single great idea embodied in the Second Law of thermodynamics.*


There is no doubt that the notion of entropy entails difficulties in understanding, but this happens because our education is based on a deterministic paradigm. Indeed, it is difficult to incorporate a clearly stochastic concept, i.e., entropy, into a deterministic mindset. The notion of order looks determinist-friendly, and its opposite, disorder, has a negative connotation in the deterministic mindset.

However, the notions of order and disorder are less appropriate and less rigorous as scientific terms, and more appropriate in describing mental states (as in Wiener’s use described above; cf. personality disorder, stress disorder, bipolar disorder, mental disorder), and even more so in describing socio-political states. The latter is manifest in the frequent use of the expressions “world order” and “new world order” in political texts, included in Google Books (see Figure 3). As a further example, the phrases “world order” and “new world order” were favorites of Kissinger, the American geopolitical consultant who served as United States Secretary of State and National Security Advisor. Specifically, they appear in 109 and 28 of his articles registered in Google Scholar, respectively [89,90], of which the article from 2009 [91] was his most popular, until recently (48 citations). However, the variant “new world order” used by Kissinger in [91] has become infamous, as has the Nazis’ “new order” after World War II (see Figure 3). Naturally, Kissinger changed it in his article from 2020 [92], which by now has become even more popular (132 citations), to “liberal world order”, using this phrase for first time but borrowing it from other authors (e.g., [93]). As will be explained in the next few lines, the latter expression is self-contradictory, or a euphemism.

Other representatives of oligarchic elites prefer the expression “global order” (also included in Figure 3). For example, in the recent book from the World Economic Forum “COVID-19: The Great Reset” [94], the latter expression appears seven times (and “global disorder” once), while “new order” and “world order” do not appear at all (except in a reference to Kissinger’s article). Note that the book invokes the “COVID-19 pandemic” (appearing 14 times) along with “climate change” (appearing 37 times), “global warming” (appearing 4 times), and “climate crisis” (appearing twice), to promote the idea of a “great reset” (appearing 13 times). The latter comprises economic reset, societal reset, geopolitical reset, environmental reset, industry and business reset, and even individual reset.

In one of the earliest critiques of the disorder interpretation of entropy, Wright [97] made a plea for moderation in the use of “intuitive qualitative ideas concerning disorder”. With a more absolute tone, Leff [98] recently stated:


*The too commonly used disorder metaphor for entropy is roundly rejected.*


In an even more recent article, Styer [99] stated:
*we cannot stop people from using the word “entropy” to mean “disorder” or “destruction” or “moral decay.” But we can warn our students that this is not the meaning of the word “entropy” in physics.*
Steyer attributes an excessive contribution to the misconception of entropy as disorder to the autobiographical book “The Education of Henry Adams” [100]. He relates that it proved to be enormously influential, as it won the 1919 Pulitzer Prize in biography, and in April 1999 was named by the Modern Library the 20th century’s best nonfiction book in English. As quoted by Steyer, Adams contrasts chaos and anarchy, and states:


*The kinetic theory of gas is an assertion of ultimate chaos. In plain words, Chaos was the law of nature; Order was the dream of man.*


This looks to be a very strong statement. Undoubtedly, elites that want to control the world have exactly this dream (cf. [101] and references above). However, this does not necessarily mean that all of humanity has the same dream as the elites. When speaking about entropy, we should have in mind that the scale is an important element, and that entropy per se, being a probabilistic concept, presupposes a macroscopic view of phenomena, rather than a focus on individuals or small subsets. If we viewed the motion of a particular die-throw, we might say that it was irregular, uncertain, unpredictable, chaotic, or random. However, macroscopization, by removing the details, may also remove irregularity. For example, the application of the principle of maximum entropy to the outcomes of a die-throw results in equal probabilities (1/6) for each outcome. This is the perfect order that can be achieved macroscopically. Likewise, as already mentioned, the maximum uncertainty in a particular water molecule’s state (in terms of position, kinetic state and phase), on a macroscopic scale results in the Clausius–Clapeyron law. Again, we have perfect order, as the accuracy of this law is so high that most people believe that it is a deterministic law.

However, if entropy is not disorder, what is it? This question is not as difficult to answer as the above discussion seems to imply. According to its standard definition, which will be repeated in Section 2.6, entropy is precisely the expected value of the minus logarithm of probability. If this sounds too difficult to interpret, an easy and accurate interpretation (again explained in Section 2.6) is that entropy is a measure of uncertainty. Hence, maximum entropy means the maximum uncertainty that is allowed in natural processes, given the constraints implied by natural laws (or human interventions). It should be stressed that, with this general definition, entropy and its maximization do not apply merely to physics—in particular to thermodynamics—but to any natural (or even uncontrolled artificial) process in which there is uncertainty that necessitates a (macroscopic) probabilistic description. This application is not meant as an “analogy” with physics. Rather, it is a formal application of the general definition of entropy, which relies on stochastics.

If “disorder” is regarded as a “bad thing” for many, the same is the case with uncertainty. The expressions “uncertainty monster” and “monster of uncertainty” appear in about 250 scholarly articles registered in Google Scholar (samples are [102,103], to mention a couple of the most cited with the word “monster” appearing in their title). However, if uncertainty is a monster, it is thanks to this monster that life is liveable and fascinating. Uncertainty is not an enemy of science or of life; rather, it is the mother of creativity and evolution. Without uncertainty, life would be a “universal boredom” (to borrow a phrase by Saridis [21] and reverse its connotation), and concepts such as hope, will (particularly, free will), freedom, expectation, optimism, etc., would hardly make sense. A technocratic system wherein an elite comprising super-experts who, using super-models, could predict the future without uncertainty would also assume full control of the society [104]. Fortunately, this will never happen because entropy, i.e., uncertainty, is a structural property of nature and life. Hence, in our view, uncertainty is neither disorder nor a “bad thing”. How could the most important law of physics (the Second Law) be a “bad thing”?

In a deterministic world view, there is no uncertainty, and there is no meaning in speaking about entropy. If there is no uncertainty, each outcome can be accurately predicted, and hence there are no options. In contrast, in an indeterministic world, there is a plurality of options. This corresponds to the Aristotelian idea of *δύναμις* (Latin: *potentia*—English: *potency* or *potentiality*). The existence of options entails that there is freedom, in the following sequence:*entropy ↔ uncertainty ↔ plurality of options ↔ freedom.*

This view, also depicted in Figure 4, is consistent with what has been vividly expressed by Brissaud [71]:


*Entropy measures freedom, and this allows a coherent interpretation of entropy formulas and of experimental facts. To associate entropy and disorder implies defining order as absence of freedom.*


### 2.4. On Negentropy

In 1921, the Swiss physicist C.-E. Guye [107] (followed by other scientists) asked the question: How is it possible to understand life, when the whole world is ruled by such a law as the second principle of thermodynamics, which points toward death and annihilation? Today it makes sense to ask: Has this question been answered by now? Or, is it still relevant, one hundred years after? As insightfully discussed by Brillouin (1949 [108]), scientists of the era wondered if there was a “life principle”, a new and unknown principle that would explain life as an entity that contrasts the second law of thermodynamics. A year after, Brillouin coined the term *negentropy* as an abbreviation of negative entropy [109]. In this, he used information theoretical concepts to express the idea that every observation in a laboratory requires the degradation of energy, and is made at the expense of a certain amount of negentropy, taken away from the surroundings.

The term “negative entropy” had earlier (in 1944) been used by Schrödinger in his famous book “What is life?” [110]. Specifically, he argued that “What an organism feeds upon is negative entropy”. At the same time, he did not mention any other “life principle” additional to the Second Law that would drive life and evolution.

There is no general agreement about the meaning of negative entropy or negentropy. Some (e.g., [111]) use them as technical terms referring to the difference between the entropy of any variable and that of a variable with normal distribution, with the same mean and variance (distance to normality). However, others, in a rather metaphysical context and assuming a non-statistical definition of negentropy (e.g., [112]), see a negentropic principle governing life, the biosphere, the economy, etc., because these convert things that have less order into things with more order.

On the other hand, Atkins [64], who, as we have seen, explains entropy as disorder, neatly remarked:

*The ceaseless decline in the quality of energy expressed by the Second Law is a spring that has driven the emergence of all the components of the current biosphere.* […] *The spring of change is aimless, purposeless corruption, yet the consequences of interconnected change are the amazingly delightful and intricate efflorescences of matter we call grass, slugs, and people.*

Apparently, if we get rid of the disorder interpretation of entropy, we may also be able to stop seeking a negentropic “life principle”, which was never found and probably will never be. For, if we see entropy as uncertainty, we also understand that life is fully consistent with entropy maximization. Human-invented steam engines (and other similar machines) increase entropy all the time, and are fully compatible with the Second law, yet they produce useful work. Likewise, the biosphere increases entropy, yet it produces interesting patterns, much more admirable than steam engines. Life generates new options and increases uncertainty [113]. Compare Earth with a lifeless planet: Where is uncertainty greater? On which of the two planets would a newspaper have more events to report every day?

### 2.5. Final Theses on Entropy

The above considerations allow us to form a logical basis for a general entropic framework, which can be applicable in many scientific fields including thermodynamics, geosciences and social sciences. This includes the following points:Entropy is a stochastic concept with a simple and general definition that will be formally stated in Section 2.6. Notably, according to its stochastic definition, entropy is a dimensionless quantity;As a stochastic concept, entropy can be interpreted as a measure of uncertainty, leaving aside the traditional but obscure and misleading “disorder” interpretation;The classical definition of thermodynamic entropy is not necessary, and it can be abandoned and replaced by the probabilistic definition;Applied in thermodynamics, entropy thus defined is the fundamental quantity, which supports the definition of all other derived ones. For example, temperature is defined as the inverse of the partial derivative of entropy with respect to internal energy. Entropy retains its dimensionless character in thermodynamics, thus rendering the kelvin an energy unit. Notably, the extended and sophisticated study of entropy in thermodynamics can serve, after the removal of the particulars pertinent to this specific field, as a paradigm for other disciplines, given that entropy is a generic concept;The entropy concept is complemented by the principle of maximum entropy, which states that entropy tends to take the maximum value that is allowed, given the available information about the system. The latter is incorporated into maximization in the form of constraints. This can be regarded both as a physical (ontological) principle obeyed by natural systems, as well as a logical (epistemological) principle applicable when making inferences about natural systems;The tendency of entropy to reach its maximum is the driving force of natural change;Life, biosphere and social processes are all consistent with the principle of maximum entropy, as they augment uncertainty. Therefore, no additional “life principle” is necessary to explain them. Changes in life and evolution are also driven by the principle of maximum entropy.

### 2.6. Mathematical Formulation

We consider a stochastic (random) variable $\underline{x}$ (notice that we underline stochastic variables to distinguish them from common variables), and we denote its distribution function (i.e., probability of non-exceedance) and its tail function (i.e., probability of exceedance), respectively, as:(1)$$F\left(x\right)≔P\left\{\underline{x}\leq x\right\},\;\bar{F}\left(x\right)=1-F\left(x\right)=P\left\{\underline{x}>x\right\}$$
where *P* denotes probability. If the variable $\underline{x}$ is discrete, i.e., it can take any of the values $x_{j},j=1,\ldots ,\Omega$, with probability
(2)$$P_{j}\equiv P\left(x_{j}\right)≔P\left\{\underline{x}=x_{j}\right\}$$
then the sequence $P_{j}$ defines its probability mass function. If the variable is continuous, i.e., it can take any real value (or a value in a subset of the real numbers), then we define the probability density function as the derivative of the distribution function:(3)$$f\left(x\right)≔\frac{dF\left(x\right)}{dx}$$
The sequence $P_{j}$ and the function $f\left(x\right)$ obey the obvious relationships:(4)$$\overset{\Omega }{\underset{j=1}{\sum }}\;P_{j}=1,\;\;\overset{\infty }{\underset{-\infty }{\int }}f\left(x\right)dx=1$$

Any deterministic function of $\underline{x}$, $g\left(\underline{x}\right)$, is a stochastic variable per se, because its argument is stochastic. The expectation of the stochastic variable $g\left(\underline{x}\right)$ is defined as
(5)$$E\left[g\left(\underline{x}\right)\right]≔\overset{\Omega }{\underset{j=1}{\sum }}g\left(x_{j}\right)\;P_{j},\;E\left[g\left(\underline{x}\right)\right]≔\overset{\infty }{\underset{-\infty }{\int }}g\left(x\right)f\left(x\right)dx$$
for a discrete and continuous stochastic variable, respectively. For $g\left(\underline{x}\right)=\underline{x}$, we get the mean of $\underline{x}$*,* as
(6)$$\mu ≔E\left[\underline{x}\right]≔\overset{\Omega }{\underset{j=1}{\sum }}x_{j}\;P_{j},\;\mu ≔E\left[\underline{x}\right]≔\overset{\Omega }{\underset{j=1}{\sum }}xf\left(x\right)dx$$
and for $g\left(\underline{x}\right)={\left(\underline{x}-\mu \right)}^{2}$, we get the variance of $\underline{x}$ as
(7)$$\begin{array}{cc} \gamma ≔E\left[{\left(\underline{x}-\mu \right)}^{2}\right] & ≔\sum_{j=1}^{\Omega }{\left(x_{j}-\mu \right)}^{2}\;P_{j},\\ & \gamma ≔E\left[{\left(\underline{x}-\mu \right)}^{2}\right]≔\overset{\infty }{\underset{-\infty }{\int }}{\left(x-\mu \right)}^{2}f\left(x\right)dx \end{array}$$
The variance is necessarily nonnegative, and its square root, $\sigma ≔\sqrt{\gamma }$, is the standard deviation. For nonnegative variables, the ratio σ/μ, termed the coefficient of variation, is a useful dimensionless index of the variability of a system.

In the above presentation of these basic probabilistic notions, we have followed Kolmogorov’s axiomatic system of probability [114,115], and we will do the same in what follows. According to this system, the definition of a stochastic variable $\underline{x}$ entails an enumeration of the basic set (the set of all possible elementary events). Hence, it reflects arbitrary choices (e.g., about units) as there are many different options for enumeration. In turn, expectations and moments depend on the option chosen. One may think of defining the function $g\left(\;\right)$ whose expectation is sought in terms of the probability per se, i.e., $g\left(\underline{x}\right)=h\left(P\left(\underline{x}\right)\right)$ for a discrete variable or $g\left(\underline{x}\right)=h\left(f\left(\underline{x}\right)\right)$ for a continuous variable, where $h\left(\;\right)$ is any specified function. Among the several choices of $h\left(\;\right)$, the most useful is the logarithmic function, which results in the definition of entropy.

The emergence of the logarithm in the definition of entropy follows some postulates originally set up by Shannon (1948). Assuming a discrete stochastic variable $\underline{x}$ with probability mass function $P_{j}\equiv P\left(x_{j}\right)$, which satisfies Equation (4), the postulates, as reformulated by Jaynes [116] (p. 347]), are:(a)It is possible to set up a numerical measure Φ of the *amount of uncertainty*, which is expressed as a real number;(b)*Φ* is a continuous function of $P_{j}$;(c)If all the $P_{j}$ are equal ($P_{j}=1/\Omega$), then Φ should be a monotonic increasing function of Ω;(d)If there is more than one way of working out the value of Φ, then we should get the same value for every possible way.

Quantification of postulate (d) is given in, among others, in Refs. [70] (p. 3) and [117], (theorem 1), and is related to the refinement of partitions to which the probabilities $P_{j}$ refer.

From these general postulates about uncertainty, a unique (within a multiplicative factor) metric Φ is derived, which serves as the definition of entropy:(8)$$\Phi \left[\underline{x}\right]≔\;E\left[-\ln\;P\left(\underline{x}\right)\right]=-\overset{\Omega }{\underset{j=1}{\sum }}P_{j}\ln P_{j}$$
While, as we have seen, in classical thermodynamics, entropy is denoted by *S* (the original symbol used by Clausius; see Section 2.1), probability texts use the symbol *H*. Here, *Φ* was preferred as a unifying symbol for information and thermodynamic entropy, under the interpretation that the two are essentially the same thing.

The extension of the above definition for the case of a continuous stochastic variable $\underline{x}$ with probability density function $f\left(x\right)$ is possible, although not contained in Shannon’s (1948) original work. This extension presents some difficulties. Specifically, if we discretize the domain of x into intervals of size δx, then (8) would give an infinite value for the entropy, as δx tends to zero (the quantity $-\ln P=-\ln\left(f\left(x\right)\;\delta x\right)$ will tend to infinity). However, if we involve a (so-called) *background measure* with density $\beta \left(x\right)$ and take the ratio $\left(f\left(x\right)\delta x\right)/\left(\beta \left(x\right)\delta x\right)=f\left(x\right)/\beta \left(x\right)$, then the logarithm of this ratio will generally converge. This allows for the definition of entropy for continuous variables as (see, e.g., [116]; p. 375, [117]):(9)$$\Phi \left[\underline{x}\right]≔E\left[-\ln\frac{f\left(\underline{x}\right)}{\beta \left(\underline{x}\right)}\right]=-\overset{\infty }{\underset{-\infty }{\int }}\ln\frac{f\left(x\right)}{\beta \left(x\right)}f\left(x\right)dx$$
The background measure density $\beta \left(x\right)$ can be any probability density, proper (with integral equal to 1, as in Equation (4)) or improper (meaning that its integral diverges). Typically, it is an (improper) Lebesgue density, i.e., a constant.

We note that most texts do not include $\beta \left(\underline{x}\right)$ in the definition (or set $\beta \left(\underline{x}\right)\equiv 1$) but in terms of physical consistency this is an error, because in order to take the logarithm of a quantity, this quantity must be dimensionless. While the probability mass $P\left(x\right)$ in a discrete variable is indeed dimensionless, the density function has units $\left[f\left(x\right)\right]=\left[x^{-1}\right]$, and therefore we need to divide it by a quantity with the same units before taking the logarithm. Even if we choose the Lebesgue measure as the background, with $\beta \left(x\right)=1/\lambda ,$ (constant), where λ is the unit used to measure x, the entropy still depends on the unit. It can easily be verified that if we measure x with two different units $\lambda_{1}$ and $\lambda_{2}$, the respective entropies $\Phi_{1}\left[\underline{x}\right]$ and $\Phi_{2}\left[\underline{x}\right]$ will differ by a constant:(10)$$\Phi_{1}\left[\underline{x}\right]-\Phi_{2}\left[\underline{x}\right]=\ln\frac{\lambda_{2}}{\lambda_{1}}$$
In other words, in contrast to the discrete variables where the entropy for a specified probability mass function is a unique number, in continuous variables, the value of entropy depends on the assumed $\beta \left(x\right)$.

Furthermore, we note that in texts that miss the background measure in the definition of entropy, the quantity that is defined in Equation (9), taken with a negative sign, is named the *relative entropy* or the *Kullback–Leibler divergence*, as it measures how the density function $f\left(x\right)$ differs from $\beta \left(x\right)$.

It can easily be seen that for both discrete and continuous variables, the entropy $\Phi \left[\underline{x}\right]$ is a dimensionless quantity. For discrete variables, it can only take nonnegative values up to a maximum value, depending on the system. For continuous variables it can be either positive or negative, depending on the assumed $\beta \left(x\right)$, ranging from −∞ to a maximum value, depending on the system and, in particular, on its constraints.

If there is no constraint in the system apart from a maximum value Ω, i.e., if the system only obeys the inequality constraint of
(11)0≤x≤Ω
then the maximization of entropy results in uniformity, i.e., $P\left(x_{j}\right)=1/\Omega$ or $f\left(x\right)=1/\Omega$. In this case, the maximum entropy is
(12)$$\Phi \left[\underline{x}\right]=\ln\Omega ,\;\;\Phi \left[\underline{x}\right]=\ln\frac{\Omega }{\lambda }$$
for the discrete and the continuous case, respectively. The former equation corresponds to Boltzmann’s original definition of entropy, a form of which has been carved on his gravestone. In the latter case, a Lebesgue background measure is assumed, i.e., $\beta \left(x\right)=1/\lambda$.

However, a system becomes more interesting when, in addition to inequality constraints such as (11), or even in the absence of them, there appear equality constraints, corresponding to the information that is known about a system represented by the variable $\underline{x}$. Their formulation is typically given in the form of expectations of one or more function $g_{i}\left(\underline{x}\right)$ for some i:(13)$$\;E\left[g_{i}\left(\underline{x}\right)\right]=\gamma_{i}\Leftrightarrow \overset{\infty }{\underset{-\infty }{\int }}g_{i}\left(x\right)f\left(x\right)dx-\gamma_{i}=0$$
As shown in [118], for any background measure $\beta \left(x\right)$, after incorporating the constraints to the entropy with Lagrange multipliers, the entropy maximizing density is:(14)$$f\left(x\right)=A\;\beta \left(x\right)\exp\left(-\underset{i}{\sum }b_{i}g_{i}\left(x\right)\right)$$
where *A* and $b_{i}$ are the parameters to be determined from the constraints of Equations (4) and (13). Once $f\left(x\right)$ is determined, the maximum entropy is calculated by Equation (9).

In closing the presentation of our general entropic framework, we return to the definition of entropy, and stress the importance of the postulate (d). This allows us to separate a whole system into partitions, requiring that, as the entropy of the whole is maximized, the partial entropy in each of the partition blocks should also be maximized. In turn, this enables a study of subsystems without necessarily considering the entire system. For example, we can study the economy of a country without considering all processes on Earth or in the universe.

## 3. Entropy Maximizing Distribution for Constrained Mean

### 3.1. Lebesgue Background Measure and the Exponential Distribution

When studying the material wealth (or income) in a certain society, current or past, we assume two characteristic quantities: the mean *μ*, which is related to the total energy available to the society [119], and the upper limit of wealth (or income) Ω, which is mainly determined by the available technology (knowhow), and can thus be called the technological upper limit. We define the ratio:(15)$$A≔\frac{\Omega }{\mu }\geq 1$$
Hence, in entropy maximization, we have an equality constraint and an inequality one, i.e.,
(16)$$\overset{\infty }{\underset{-\infty }{\int }}xf\left(x\right)dx=\mu ,\;\;0\leq x\leq A\mu$$

The probability that maximizes entropy is determined from the general solution (14). Assuming a Lebesgue background measure with $\beta \left(x\right)=1/\lambda$, with *λ* being a monetary unit (e.g., *λ* = USD 1), after algebraic manipulations, we find the entropy maximizing probability density to be:(17)$$f\left(x\right)=\frac{b\;e^{-bx/\mu }}{\mu \left(1-e^{-bA}\right)}$$
which is a (doubly) bounded exponential distribution (or anti-exponential if b<0). The value of b depends on *A* and is the solution to the implicit equation:(18)$$\frac{1}{b}-\frac{A}{e^{Ab}-1}=1$$
which renders b a function of A, $b\left(A\right)$, albeit implicitly determined. The entropy is then found to be:(19)$$\Phi \left[\underline{x}\right]=b\left(A\right)+\ln\left(\frac{\mu }{\lambda }\right)+\ln\left(\frac{A}{\left(A-1\right)\;b\left(A\right)+1}\right)≕\Phi \left(\mu ,A\right)$$

Interesting special cases of the general solution of (17) are encountered for A=1 (an impulse function with all mass concentrated at x=μ, representing certainty), A=2 (the uniform distribution) and A→∞ (the unbounded exponential distribution). Their particular characteristics are given in Table 1 as functions of *μ* and in Table 2 as functions of Ω.

Accurate solutions of Equation (18) can be directly calculated in terms of the auxiliary variable:(20)$$c≔e^{Ab}$$
Starting with a known c, the exact solution is readily found by
(21)$$A={\left(\frac{1}{\ln c}-\frac{1}{c-1}\right)}^{-1},\;b=1-\frac{\ln c}{c-1}$$
where for c≥e we get A≥2,b≥0, and for c≤e we get A≤2,b≤0. Two sample solutions for $c=2^{-3}$ and $c=2^{3}$ are also shown in Table 1 and Table 2.

In a typical case where c is unknown, we need to solve Equation (18) numerically, which is not too difficult. Alternatively, we can use the following very good approximation for A≥2:(22)$$b=1-\left(1-1.37e^{-0.89}\right)\;e^{2\left(2-A\right)}-1.37{\;e}^{0.89\left(1-A\right)}$$
Furthermore, a very good approximation for entropy $\Phi \left[\underline{x}\right]$, which does not contain *b* at all, is:(23)$$\Phi \left[\underline{x}\right]=1+\ln\left(\frac{\mu }{\lambda }\right)-\left(1-\ln2-0.373e^{-1}\right)\;e^{2\left(2-A\right)}-0.373{\;e}^{\left(1-A\right)}$$
Approximations (22) and (23) were found via an extensive numerical investigation and are optimized for A≥2. For A<2, we can exploit the following symmetry relationships:(24)$$b\left(A\right)=-\frac{1}{A-1}b\left(\frac{A}{A-1}\right),\;\;\Phi \left(\mu ,A\right)=\Phi \left(\left(A-1\right)\mu ,\frac{A}{A-1}\right)$$
where it is readily seen that if A<2, then $A/\left(A-1\right)>2$.

In addition to their tabulated form (Table 1), the characteristic density functions are also depicted in graphical form in Figure 5. Likewise, the density functions of Table 2 are depicted in Figure 6. Furthermore, Figure 7a shows the achieved maximum entropy for a constant mean μ=1, as a function of the technological limit Ω. For small and moderate values of the technological limit, the entropy is an increasing function of Ω, but beyond Ω≈5, it reaches a virtually constant value. Likewise, Figure 7b shows the maximum entropy for a constant technological limit Ω=2.958 (this value corresponds to case #4 of Table 2) as a function of the mean *μ*. Initially, for small *μ*, the entropy increases; it takes a maximum value for μ=Ω/2 and then decreases in a symmetric pattern. However, in the case that technology offers unlimited opportunities (infinite technological limit), as also depicted in Figure 7b, the increase in entropy with μ is continuous. We can thus say in conclusion that for Ω>2μ, the entropy increases both with the mean *μ* and the technological limit Ω. In this respect, entropy constitutes a measure of society’s wealth (see also [119]).

One could say that the mean *μ* is more representative, as a measure of wealth, than the entropy Φ. We do not make a substantial objection to this. However, we prefer to use entropy for two reasons: (a) because it is connected with the logarithm of the mean, and intuitively this is a better quantification of the wealth; (b) because it quantifies the options, and as seen in the motto at the beginning of the paper, the quantification of options is more pertinent as a measure of average wealth.

Apparently, life offers much more options than material wealth, and one could choose to pursue different opportunities (e.g., intellectual), snubbing material wealth, as formulated by Bias of Priene in the second motto at the beginning of the paper. Certainly, the focus of the paper is on material wealth, but we should keep in mind that seeking material wealth is just one of the options (for example, many, including us, would not like to exchange their lives with those of any of the persons whose income is discussed in Section 3.3 below).

While for a constant background density equal to the inverse of the monetary unit (i.e., 1/λ with λ equal, e.g., to USD 1) the entropy provides a measure of society’s wealth (even if x expresses income), if we change the background measure to the value 1/μ, where μ is the mean income, the resulting entropy is a measure of inequality. Calling the latter quantity *standardized entropy* and denoting it as $\Phi_{\mu }\left[\underline{x}\right]$, from Equation (10) we get
(25)$$\Phi_{\mu }\left[\underline{x}\right]=\Phi \left[\underline{x}\right]-\ln\frac{\mu }{\lambda }$$
This has recently been introduced as an index of inequality by Sargentis et al. [119] (albeit denoted as $\Delta \Phi \left[\underline{x}\right]).\;\mathrm{The}\;\mathrm{quantity}\;\Phi_{\mu }\left[\underline{x}\right]$ cannot exceed a maximum value of 1, corresponding to an exponential distribution. A value smaller than 1 usually indicates less inequality with respect to the exponential distribution. However, as we will see in Section 3.2, there are cases where it indicates higher inequality, and hence the value of $\Phi_{\mu }\left[\underline{x}\right]$ should be accompanied with a second inequality index in order to decide whether the inequality is lower or higher. As we will see below, a simple and appropriate additional index is the coefficient of variation σ/μ.

One issue that needs to be discussed here is whether or not the mean μ used in the above formulation indeed represents an essential constraint, and should be used in entropy maximization. In physics, the related constraint is imposed by natural laws of conservation—most prominently, energy conservation. Is there sense in imposing a constrained mean when using economic variables, such as income? Related to this question is Tusset’s [120] note:


*However, it is the relationship between thermodynamics and economics (hardly a new topic), with its burden of “entropy” and information, that remains at the heart of any econophysics view of production. In a nutshell, the point is: thermodynamics implies the conservation of energy, a principle that so far has not been confirmed in economic processes.*


This question is also thoroughly discussed by Yakovenko and Rosser [54]. In our view, even in physics where energy is definitely conserved, this conservation applies to isolated systems, and certainly not to open ones. Yet energy constraint is relevant even for configurations that form open systems, provided that we deal with specified periods of time in which energy changes can be neglected. Likewise, in economic processes in general, there is no conservation of a quantity that could be regarded as a substitute of energy, such as money. However, we can assume that in a system with large spatial extent (e.g., a country or the globe), such changes are slow, and for a relatively small period of time (e.g., a year, which is the basis of most statistical economic data), the mean can be regarded as constant. Apparently, though, if we consider the entirety of history, the mean is evolving toward higher values. We will illustrate this idea in Section 4.1, examining hypothetical scenarios of historical evolution, each one corresponding to a period with a constant mean, and where entropy maximization determines the distribution throughout the population of the values that shape this mean.

### 3.2. Hyperbolic Background Measure and the Pareto Distribution

Coming again to the quantification of material wealth, we recall that the obtained density function and maximum entropy would be different if we choose a different background measure. In particular, if we choose a hyperbolic background measure [80,118], i.e.,
(26)$$\beta_{H}\left(x\right)=\frac{1}{\lambda +x}$$
leaving x unbounded, and constrain a generalized mean, consistent with the chosen background measure, then the entropy maximizing distribution comes to be of Pareto type, with density
(27)$$f\left(x\right)=\frac{1}{\lambda_{1}}\;{\left(1+\xi \frac{x}{\lambda_{1}}\right)}^{-\frac{1}{\xi }-1},\;x\geq 0$$
where $\xi ≔1/b_{1}$ (the parameter in Equation (14), known as the tail index) and $\lambda_{1}≔\lambda \xi$. It can be seen (by taking the limit) that, as ξ→0, the Pareto density (Equation (27)) tends to the unbounded exponential one (case #5 in Table 1), with $\mu =\lambda_{1}$.

Equation (27) could also be derived in a different manner by using a generalized definition of entropy, the so-called Havrda–Charvát–Tsallis entropy [121,122]. This has been used in several econophysics studies (e.g., [123,124]). However, we contend that the derivation using the classical (Boltzmann–Gibbs–Shannon) definition (and an appropriate, non-Lebesgue background measure) is more natural and advantageous as it satisfies Shannon’s postulates and retains the properties resulting from these postulates. In contrast, the Havrda–Charvát–Tsallis entropy does not satisfy postulate (d). (About the importance of that postulate, see the ultimate paragraph of Section 2.6.)

We note that the density in Equation (27) is often called the Pareto Type II or Lomax distribution, while the name Pareto distribution (more precisely, Pareto Type I distribution) is used for a case wherein the 1 in the parentheses in Equation (27) is neglected; in that case, x cannot take a value smaller than $\lambda_{1}/\xi$. The differences in the two cases are negligible for a large x.

The distribution, elsewhere known as the “Pareto’s law”, is named after Vilfredo Pareto, who first proposed it in 1896 while analyzing data on the distribution of wealth and fitting a straight line on the logarithm of the number of people, $N_{x}$, whose net worth or income exceeds x, and the logarithm of x [125]. A detailed historical account of his discovery is given by Mornati [126], where it can be seen that Pareto developed both types I and II of the distribution. In fact, it was not the power–law behavior of the distribution that impressed Pareto. As evident from his celebrated book *Manual of Political Economy*, first published in 1906, he was rather puzzled by its asymmetric shape, which is also shared by the exponential distribution. Specifically, he drew a qualitative shape of the distribution in Figure 54 of his book, and remarked [127] (p. 195):

*The shape of the curve* […] *of Figure 54, which is derived from statistics, does not correspond by any means to the curve of errors* [i.e., the normal distribution]*, i.e., to the shape the curve would have if the acquisition and preservation of wealth depended only on chance. Moreover, statistics reveal that the curve* […] *varies very little in time and space; different nations at different times have very similar curves. There is thus a remarkable stability in the shape of this curve.* […] *There is a certain minimum income* […] *below which men cannot descend without dying of poverty and hunger.*

It is remarkable that Pareto regarded that “chance” is only connected with normal distribution—an idea that is not consistent with reality. If we accept that what Pareto regarded as “chance” can be represented by the principle of maximum entropy, then it is true that there is consistency between the latter principle and the normal distribution. However, this occurs when the second moment of $\underline{x}$ is constrained to a constant value (as, e.g., in the kinetic energy of a number of molecules), and such a constraint is certainly not applicable to income. Instead, as we have seen, reasonable constraints for the income necessarily result in asymmetric distributions, of either exponential or Pareto type. In this respect, the Pareto distribution is applied, and has become popular in financial analysis, in either its original form or after generalization [128].

Coming back to the mathematical details, the mean and coefficient of variation of the Pareto distribution are
(28)$$\mu =\frac{\lambda_{1}}{1-\xi }=\frac{\lambda \xi }{1-\xi },\;\frac{\sigma }{\mu }=\frac{1}{\sqrt{1-2\xi }}$$
Assuming that both have finite values, a restriction is imposed for ξ, i.e., 0≤ξ<1/2. The maximized entropy for the hyperbolic background measure is
(29)$$\Phi_{H}=1+\ln\xi =1-\ln\left(1+\lambda /\mu \right)$$
while if, for the sake of compatibility, we also calculate the entropy for the Lebesgue measure $\beta \left(x\right)=1/\lambda$, this entropy is
(30)$$\Phi =1+\xi +\ln\xi =1+\frac{1}{1+\lambda /\mu }-\ln\left(1+\lambda /\mu \right)$$
The proofs are omitted for both cases. It can also be shown (again the proof is omitted) that the following inequality holds true:(31)$$\Phi_{H}\leq \Phi \leq 1+\ln\left(\frac{\mu }{\lambda }\right)$$
where the rightmost term is the entropy of the unbounded exponential distribution. The three quantities become equal, as μ/λ→0 (or, equivalently, as ξ→0).

It is relevant here to discuss the so-called “Pareto principle” or the “80/20 rule” referred to in economic papers (e.g., [129]), suggesting that about 80% of wealth is concentrated in about 20% of any population. The mathematics for this principle is the following. Let $\mu_{x\geq c}≔\int_{c}^{\infty }xf\left(x\right)dx$ be the mean of $\underline{x}$ conditional on $\underline{x}>c$. The “principle” states that for $\mu_{x\geq c}/\mu =0.8$, $\bar{F}\left(c\right)=0.2$.

It can be shown that for the Pareto distribution, the following relationship holds true:(32)$$\frac{\mu_{x\geq c}}{\mu }=\frac{\left(1+\left(\bar{F}{\left(c\right)}^{-\xi }-1\right)/\xi \right)\bar{F}\left(c\right)}{1-\xi }$$
and it can be readily verified that the condition $\mu_{x\geq c}/\mu =0.8,\;\bar{F}\left(c\right)=0.2$ is met when ξ=0.253. For ξ→0, the Pareto distribution reduces to the exponential distribution, and Equation (32) reduces to
(33)$$\frac{\mu_{x\geq c}}{\mu }=\left(1-\ln\bar{F}\left(c\right)\right)\bar{F}\left(c\right)$$
From the latter, it is easy to verify that the condition $\mu_{x\geq c}/\mu =0.8$ is satisfied when $\bar{F}\left(c\right)=0.439$, that is, substantially higher than 0.2.

When the articles refer to the “Pareto principle”, they usually cite Pareto’s Manual [127]. However, our own research did not show that the “principle” is referred to or implied by his book. On the contrary, it appears that this name was accidentally given in 1951 by Juran [130], as explained by himself in his paper entitled “The non-Pareto principle; mea culpa”, where he claims that it was himself who introduced the principle, also using the name “vital few and trivial many” [131]. For these reasons, in the following we will call this “principle” the “80/20 rule”, and we will see that real-world data on income do not satisfy it.

A graphic comparison of the Pareto density with the exponential one is given in Figure 8. In addition, the two-parameter gamma density, the behavior of which is opposite to Pareto, is also plotted in the figure. The gamma density, with a chosen shape parameter of 2, is $f\left(x\right)=xe^{-x/a}/a^{2}$, and has a mean μ=2a, a coefficient of variation 0.5 and entropy
(34)$$\Phi =1+\gamma +\ln a/\lambda =1+\gamma -\ln2+\ln\left(\mu /\lambda \right)=0.884+\ln\left(\mu /\lambda \right)$$
where γ=0.5772… is the Euler’s constant. For the comparison, we use the Lebesgue measure $\beta \left(x\right)=1/\lambda$. In all three distributions, the domain of x is $\left[0,\infty \right)$ and the mean is μ=λ=1. The entropy maximizing distribution is exponential, which gives entropy Φ=1. The gamma distribution has a scale parameter a=0.5 and entropy Φ=0.884<1. The tail index of the Pareto distribution was chosen so that it would have the same entropy with a gamma distribution Φ=0.884 and mean μ=1; these are obtained when ξ=0.408 and $\lambda_{1}=0.592$.

While the Gamma and the Pareto distributions have the same mean and the same entropy, they have different behaviors. The former favors (i.e., increases the frequency of) the moderate values of *x* (i.e., populating the “middle class” more), while the latter favors the extremes (diminishing the presence of the “middle class”, and populating more the “poor” and the “very rich”). These differences cannot be captured by entropy alone, as the entropy in the two cases is the same. One could think of using the concept of divergence from the exponential distribution, which is easy, as it is only required to set the background density as equal to that of the exponential distribution, i.e., $\beta \left(x\right)=\exp\left(-x\right)$ (and change the sign in the right-hand side of Equation (9)). Again, though, this does not help, as it yields almost the same divergence: 0.116 and 0.111 for the gamma and the Pareto case, respectively.

Thus, to quantify the differences in the two cases, we have to use a different metric, and the simplest is the coefficient of variation, whose values are 0.5 and 2.325 for the gamma and the Pareto cases, respectively.

Notice in Figure 8 that in the left panel the vertical axis is logarithmic, while in the right panel both axes are logarithmic. The left is better for visualizing the exponential tail of a distribution, which appears as a straight line for large *x*. This is the case for both the exponential and the gamma distribution, with their slopes being −1 and −2, respectively. The right panel better visualizes the power–law tail of a distribution, as in the Pareto case, which appears also as a straight line for large *x* with constant slope −1−1/ξ=−3.453.

In natural (e.g., geophysical) processes, both exponential and power-type tails appear, which means that the appropriate background measure, Lebesgue or hyperbolic, may differ in entropy maximization for different processes [118]. In socioeconomic processes, the question of what is the appropriate measure has not been explored. However, as already mentioned, the emergence of the Pareto distribution has been justified by using Havrda–Charvát–Tsallis entropy. In addition, as already described in detail and reflected in its name, the Pareto distribution stemmed from the field of economics. As already mentioned, since its introduction, a plethora of empirical investigations have found the Pareto distribution to be consistent with income data. To focus on the most recent related publication, Néda et al. [132] used income data from Japan (2015), USA (2013), Russia (2015), Australia (2011), Finland (2017), Hungary (2015) and a county in Romania (2005), fitting a simple and elegant model to them, and identified a beta prime distribution, which has a Pareto tail with tail index ξ=1/3. The same research team, in a follow-up publication [133], examined the wealth data for the USA and Russia and again reported a Pareto tail.

However, other studies have criticized the appropriateness of the Pareto distribution and pointed to an exponential distribution in income. Thus, Drăgulescu and Yakovenko [134] stated:

*The study of income distribution has a long history. Pareto* [127] *proposed in 1897 that income distribution obeys a universal power law valid for all times and countries. Subsequent studies have often disputed this conjecture. In 1935, Shirras* [135] *concluded: “There is indeed no Pareto Law. It is time it should be entirely discarded in studies on distribution”. Mandelbrot* [136] *proposed a “weak Pareto law” applicable only asymptotically to the high incomes. In such a form, Pareto’s proposal is useless for describing the great majority of the population.*

Bypassing the fact that this quotation, according to the above overview, severely miscites Pareto, we highlight the authors’ conclusion that their analyses “*demonstrate that the exponential law* […] *fits the individual income distribution very well*”. The title of their paper, “Evidence for the exponential distribution of income in the USA”, reflects this conclusion.

Based on the above overview, one may assume that both exponential and Pareto tails materialize in income distribution, depending on the country and time period studied. If the income distribution is consistently exponential for a long period of time, then one may infer, from the following line of thought, that the wealth (or net worth) distribution will also have an exponential tail. One fact that distinguishes exponential from Pareto distribution tails is that in the former, all moments exist, while in the latter, moments of order >1/ξ are infinite. If all moments exist in the income process, then the same will be the case for savings (smaller than income) and any linear transformation thereof. Hence, the moments of the wealth process (linear transformation, e.g., aggregation in time, of the savings process) will exist, which implies an exponential tail. On the other hand, if there are subperiods of income with a Pareto distribution tail, then the distribution tail of wealth will also be Pareto.

Interestingly, Chakrabarti et al. [53] (p. 45) characterized a society with exponential wealth distribution as “super-fair”, and one with Pareto distribution “fair” or “unfair”, depending on the tail index and other parameters. Biró and Néda [124], referring to income or wealth distributions, characterized exponential distribution as “natural”, and Pareto distribution as “capitalism”, and they provide some additional cases (“communism”, “communism++”, “eco-window”). Here we adopt the name “natural” for the exponential distribution, because it corresponds to the Lebesgue background measure where the distance between two values equals their difference (see discussion about the distance in [80,118]).

In Section 3.3 and Section 4.4, we will investigate income distribution independently of earlier studies using modern real-world data.

### 3.3. Empirical Investigation

The most informative evidence regarding the type of the distribution, and thus the appropriate background measure, is obtained by studying the distribution tail. To study the tail, we do not need to examine the entire population, i.e., the entire range of the variable x. It suffices to examine the behavior above a certain threshold $x_{0}$, and in particular the conditional tail function:(35)$$\bar{F}(x|\underline{x}>x_{0})=P\left\{\underline{x}>x|\underline{x}>x_{0}\right\}=\frac{\bar{F}\left(x\right)}{\bar{F}\left(x_{0}\right)},\;x\geq x_{0}$$

An important property of both the exponential and the Pareto distribution (not shared with other common distributions) is that, if the variable is shifted by $x_{0}$, i.e., $\underline{y}≔\underline{x}-x_{0}$, then the distribution is preserved, with only the scale parameter changing. This implies that the coefficient of variation of $\underline{x}-x_{0}$ for the values of $\underline{x}$ exceeding $x_{0}$ is the same as that of $\underline{x}$ for the entire population.

In order to empirically study the tail of income distribution, we used data on the net worth of the richest people in the world (billionaires), and the evolution thereof. We located the database referenced in the Forbes list [137] for the years 1996 to 2018. We evaluated these data with the Wayback Machine [138], and we found that amendments were needed for the years 1997, 2014 and 2015, which we performed. We noted that the Forbes data for the wealth of the 400 richest people in the United States for the earlier period 1988–2003 were already analyzed in another publication [139]. Here, we study the annual income (rather than the wealth) of the world’s richest for a longer and more recent period.

For the years 2019, 2020 and 2021, we retrieved data from Bloomberg using again the Wayback Machine [140]. Specifically, the dates we located with the Wayback Machine were 8 March 2019, 2 January 2020 (just before outbreak of the COVID pandemic) and 8 June 2021; these days allow us to study how the pandemic influenced the wealth of the richest people.

Each year’s list contains a varying number of billionaires, with an average number of 860. The dataset of all years contains about 5000 names of billionaires. Many of them appear in the list for several years. By subtracting the total net worth of a person for each year from that of the previous year, we found an approximation of the person’s annual net income (as a lower bound). Then, we sorted the list of net income in each year into decreasing order, and we took the highest 100 of them in each year, setting the threshold $x_{0}$ as equal to the annual net income of the 100th person in each year’s list. We further processed those years that contained at least 100 persons with positive annual net income—a total of 19 years.

Denoting $x_{\left(i:n\right)}$ as the net income of the person ranked i in a list of n (i=1,…,n, where in our case n=100), the unbiased estimate of the probability of exceedance of the value $x_{\left(i:n\right)}$ is [118]:(36)$$\hat{\bar{F}}\left(x_{\left(i:n\right)}|\underline{x}>x_{0}\right)=\frac{i}{n+1}$$

The sequences of $\hat{\bar{F}}\left(x_{\left(i:n\right)}|\underline{x}>x_{0}\right)$ for each year are depicted in Figure 9. In the left panel, in which the horizontal axis is linear and the vertical logarithmic, the exponential tails appear as straight lines. In contrast, in the right panel, in which both axes are logarithmic, the power–law (Pareto) tails appear as straight lines.

Inspection in both panels shows that both exponential and power–law tails appear, with the former being more common than the latter. Characteristically, in the year with the lowest earnings, 2002, the tail is clearly exponential, while in the year with the highest earnings, 2021, the tail is clearly of power–law type. It is relevant to note that 2021, which was the most anomalous as the COVID-19 pandemic negatively affected the economy globally, also brought the highest profits ever recorded to the world’s richest. This is also seen in Figure 10, where the average of the 100 highest incomes in 2021 is unprecedented, and several times higher than the average of the previous years.

Figure 10 also depicts the evolution of the coefficient of variation of the high incomes through the years with available data. We observe that the coefficient is consistently higher than unity, the value corresponding to the exponential distribution, but in most years the deviation from 1 is small. This justifies, at least as an approximation, the assumption of the exponential distribution as the “natural” norm, with the Pareto distribution being more common in anomalous periods. Put differently, the evidence from the data does not exclude the hypothesis of an exponential distribution, and hence of entropy maximization with a Lebesgue background measure. Therefore, we will use this hypothesis as the main explanatory tool for what the “natural” (and hence sustainable) tendency in the distribution of wealth is. We will provide additional evidence on the plausibility of the hypothesis in Section 4, where we will also discuss the forces that lead to deviation from the “natural” tendency.

## 4. Application to Societies’ Income Distribution

### 4.1. From the Ancient Classless Society to Modern Stratified Societies

As suggested in the recent paper by Sargentis et al. [119], in prehistoric societies wealth could be measured in terms of available energy per person. Following this idea, we identify the income x, and set x=0 to represent the energy that will cover the most basic bodily energy needs (through food intake), thus keeping a human alive. Here it is relevant to note that, while energy in a closed natural system is conserved, the energy available to human individuals or societies varies, and has substantially increased with the development of civilization.

With absence of technology (i.e., with a technological limit Ω=0), the value x=0 is the only option, and the entropy is −∞. The singular allowed option and the −∞ value of entropy, as depicted in Figure 11a, signifies a classless society. Most probably, this corresponds to a so-called hunter–gatherer society [1], admired in Marxist literature as a form of ancient communal ownership [141] (p. 44) and also aspired to in a modern form.

The notion remains popular even today (Figure 12) and, strikingly, it has been regarded as a basis for real personal freedom [141,142], despite corresponding to an entropy of minus infinity. Apparently, Marx and Engels were faithful to the deterministic scientific paradigm of their era, which they attempted to transplant into history and sociology. They could not have been aware of the modern concept of entropy. The popularity of their ideas even today [2,3] reflects the fact that the deterministic paradigm remains quite strong. In it, entropy has no place, let alone in its connection with freedom. It is relevant to quote here the seminal book by Piketty, *Capital in the Twenty-first Century* [143], whose first conclusion is that “one should be wary of any economic determinism in regard to inequalities of wealth and income”. He also notes that “Modern economic growth and the diffusion of knowledge have made it possible to avoid the Marxist apocalypse but have not modified the deep structures of capital and inequality”.

Undoubtedly, primitive societies developed some technologies in terms of stone tools, and therefore the technological limit Ω=0 depicted in Figure 11a is merely a simplification. Nonetheless, the technological revolutions that really advanced the technological limit to a higher order of magnitude were the domestication of animals (allowing the use of energy additional to that of human muscles) and the invention of agriculture. Before these developments, there could not be any storage of goods, which is a necessary condition for the very notion of wealth. At later stages, as knowledge was developed, the technological limit Ω was increased. The mean wealth μ also increased. It is plausible to assume that the rate of increase of μ always followed that of Ω. Based on this assumption, the remaining panels of Figure 11 have been constructed to represent various phases of human wealth development.

For example, at some phase of the development, the mean wealth μ was half the technological limit Ω (Figure 11b). At this phase, entropy maximization suggests that the wealth was uniformly distributed among people. Uniformity means that being poor and being rich were equally probable—not that the wealth is equally distributed among people. When the technological limit increased to more than twice the mean wealth, the distribution became (bounded) exponential. This means that the poor were more and the rich fewer in number—but they were richer than before (Figure 11c–e). However, careful inspection of the graphs shows that it is not only the richest who became richer as technology evolved. The poor also became fewer in number, and the curves $f\left(x\right)$ moved to the right, i.e., everybody, on average, became richer thanks to technological evolution. At some point in evolution, when the technological limit became very high (Figure 11e), its effect on wealth distribution became negligible. This also applies to modern societies, in which we can totally neglect this effect, replacing Figure 11e (μ=2,Ω=10) with Figure 11f (μ=2,Ω=∞). This is consistent with what we have already discussed in Figure 7a.

### 4.2. The Elites’ Role

It is reasonable to assume that the economic elites pursue a greater share of the community’s wealth. In this respect, their function can be twofold. On the one hand, they advance both the technological limit and the average wealth. On the other hand, they tend to modify the distribution of income from exponential to Pareto (Figure 8), thus increasing the number of poor and diminishing the middle class for their own benefit [3]. As has been already discussed in Section 3.3, a persuasive illustration of this is the super profit of economic elites during the recent anomalous period of the pandemic.

The means by which the elites increase their profits certainly include political power, and more recently, an attitude of world control [101]. Their endeavours are made more efficient and acceptable within the society by several means, such as by overstating existing or non-existing threats, and then by presenting themselves as philanthropists (e.g., by funding nongovernmental organizations dealing with these threats) and world saviors [94] (see also [144]). Apparently, if they succeed in controlling the world, this will decrease entropy and hence delimit freedom. In turn, this will lead to decadence, the signs of which are already visible in the Western world (cf. [145]).

### 4.3. Income Redistribution in Organized Societies

One of the important roles of a state in an organized society is the redistribution of income and wealth through their transferal from some individuals to others by means of several mechanisms, such as taxation, public services, land reform, monetary policies, and others. Such means contrast those of the elites, and aim to reduce poverty and social inequality. Here, we examine one of the mechanisms, i.e., taxation, by means of a simplified toy model example, which illustrates how redistribution affects entropy maximizing exponential distribution.

In our toy model, we assume that the original income $\underline{x}$ follows the exponential distribution with parameter λ (equal to the mean μ), and that the tax rate p increases with the original income, according to the function:(37)$$p\left(x\right)=\left\{\begin{array}{ll} 0 & x\leq x_{0}\\ p_{u} & x\geq x_{u}\\ p_{u}\frac{x-x_{0}}{x_{u}-x_{0}}, & \mathrm{otherwise} \end{array}\right.$$
where $x_{0}$ is a low value of income, denoting the starting point of taxation, and $x_{u}$ is a high value of income, beyond which the tax rate takes a constant value $p_{u}$. The tax amount will be $\underline{y}=p\left(\underline{x}\right)\underline{x}$, the income minus tax $\underline{w}=\underline{x}-\underline{y}=\left(1-p\left(\underline{x}\right)\right)\underline{x}$ and the redistributed income $\underline{z}=\underline{w}+a\mu_{y}=\left(1-p\left(\underline{x}\right)\right)\underline{x}+a\mu_{y},\;\mathrm{where}\;\mu_{y}≔E\left[\underline{y}\right]$ is the average tax and a<1 is the tax fraction that is returned to all people at equal shares. In order for the relationship of $\underline{w}$ and $\underline{x}$ to be a monotonically increasing function, the following inequalities should hold:(38)$$p_{u}<0.5,\;x_{u}>\frac{1-p_{u}}{1-2p_{u}}x_{0}$$

Using probabilistic algebra on these variables, we find that the probability density of $\underline{w}$ is:(39)$$f_{w}\left(w\right)=\left\{\begin{array}{ll} \frac{1}{\lambda }\exp\left(-\frac{w}{\lambda }\right), & 0\leq w\leq x_{0}\\ \frac{x_{u}-x_{0}}{\lambda \sqrt{D\left(w\right)}}\exp\left(-\frac{x_{u}-\left(1-p_{u}\right)x_{0}-\sqrt{D\left(w\right)}}{2\lambda p_{u}}\right), & x_{0}\leq w\leq w_{u}\\ \frac{1}{\lambda \left(1-p_{u}\right)}\exp\left(-\frac{w}{\lambda \left(1-p_{u}\right)}\right), & w\geq w_{u} \end{array}\right.$$
where
(40)$$w_{u}≔\left(1-p_{u}\right)x_{u},\;\;D\left(w\right)≔{\left(x_{u}-\left(1-p_{u}\right)x_{0}\right)}^{2}-4\left(x_{u}-x_{0}\right)p_{u}w$$
From $f_{w}\left(w\right)$ we can find the mean values:(41)$$\mu_{w}=\lambda -\frac{\lambda p_{u}}{x_{u}-x_{0}}\left(e^{-\frac{x_{0}}{\lambda }}\left(x_{0}+2\lambda \right)-e^{-\frac{x_{u}}{\lambda }}\left(x_{u}+2\lambda \right)\right),\;\mu_{y}=\lambda -\mu_{w}$$
Finally, the probability density of the final income $\underline{z}$ will be
(42)$$f_{z}\left(z\right)=f_{w}\left(z-a\mu_{y}\right)$$

Now we apply the toy model, assuming $\lambda =\mu =1,\;x_{0}=0.2,x_{u}=1.4,p_{u}=0.4,a=0.5$, resulting in $\mu_{y}=0.32,\;a\mu_{y}=0.16$. Figure 13 shows the variation in the tax, the income minus tax, and the redistributed income vs. the original income. Notice that for small incomes, the tax is zero, and thus the income minus tax equals the original income, while for larger incomes it equals 60% of the original income (Figure 13, right).

Figure 14 depicts the probability density of the redistributed income in comparison to that of the original income. The differences are that in the redistributed income: (a) poverty below the level $a\mu_{y}=0.16$ is eliminated; (b) the middle class, corresponding to incomes up to the mean, is more populated and amplified; (c) the rich lose income. As a result of this, both entropy and the coefficient of variation have been reduced in the redistributed income; the former from Φ=1 of the exponential distribution to Φ=0.55 (or $\Phi_{\mu }$ from 1 to 0.73), and the latter from σ/μ=1 to σ/μ=0.57.

### 4.4. Empirical Investigation

The empirical investigation in this section provides a comparison of the theoretical framework developed with the real-world data. Sargentis et al. [119] made similar comparisons, also intercomparing with the Lorenz curve [146,147,148] and the Gini coefficient [149,150,151], which are more standard measures of income distribution and socio-economic inequality [152,153,154,155]. In their comparisons, they used data given in tenths of the share of people from the lowest to the highest income versus the share of income earned. The partitioning into tenths is a standard form of income data structuring that is offered in relevant databases, but it fully hides the behavior of the tail, which, as we have seen, is extremely important for understanding the structural characteristics of the economy and for quantifying inequality. We will provide additional evidence for this importance in this section.

For our purposes, we searched for data given at a higher resolution than tenths of people’s income, and we found such data for USA and Sweden. Even in this case, the information about the tail (the very rich people) is missing, as the data values end at a specified level c, with the last bunch of data given as “c and over”. It is thus crucial to find a way to extrapolate the distribution function beyond c and estimate expectations based on this extrapolation.

It is consistent with our theoretical framework to assume that beyond c, the following approximation is suitable:(43)$$f\left(x\right)=e^{-x/k+b}\Rightarrow \;\bar{F}\left(x\right)=ke^{-x/k+b}=kf\left(x\right),\;\;\;x\geq c$$
where b and k are parameters to be estimated. The expectation of any function $g\left(\underline{x}\right)$ can be calculated as
(44)$$\begin{array}{c} E\left[g\left(\underline{x}\right)\right]≔\overset{\infty }{\underset{0}{\int }}g\left(x\right)f\left(x\right)dx=A^{g}+B^{g},\\ \;A^{g}≔\overset{c}{\underset{0}{\int }}g\left(x\right)f\left(x\right)dx,\;\;\;\;B^{g}≔\overset{\infty }{\underset{c}{\int }}g\left(x\right)f\left(x\right)dx \end{array}$$

The quantity $A^{g}$ can be directly estimated from the available data, by approximating the integral with a sum. Assuming that the data are given in terms of the number of persons $N_{i}$ with income between levels $x_{i-1}$ and $x_{i}$, with i=1,…n and $x_{n}\equiv c$, we have:(45)$${\hat{A}}^{g}=\overset{n}{\underset{i=1}{\sum }}g\left(\frac{x_{i-1}+x_{i}}{2}\right){\hat{f}}_{i}\;\left(x_{i}-x_{i-1}\right),\;{\hat{f}}_{i}=\frac{N_{i}}{N\left(x_{i}-x_{i-1}\right)},\;N=\overset{n}{\underset{i=1}{\sum }}N_{i}+N_{c}$$
where $N_{c}$ is the number of people with income >c.

The quantity $B^{g}$ is estimated from the approximation (43). For the moment of order p of the distribution, we have:(46)$$B_{p}≔\overset{\infty }{\underset{c}{\int }}x^{p}e^{-ax+b}dx=\frac{e^{b}}{k^{1+p}}\Gamma \left(1+p,\frac{c}{k}\right)$$
In particular, for p=0,1,2, we have
(47)$$B_{0}=ke^{b-c/k},\;B_{1}=B_{0}\left(c+k\right),\;B_{2}=B_{0}\left(k^{2}+{\left(c+k\right)}^{2}\right)$$
Since both $f\left(c\right)$ and $\bar{F}\left(c\right)$ can be directly estimated from the data as $\hat{f}\left(c\right)\equiv {\hat{f}}_{n}$ and $\hat{\bar{F}}\left(c\right)=N_{c}/N$, we have
(48)$$\hat{f}\left(c\right)=e^{-c/k+b},\;{\hat{B}}_{0}=\hat{\bar{F}}\left(c\right)=k\hat{f}\left(c\right)$$
and by solving these equations we find the unknown parameters, as
(49)$$\hat{k}=\frac{\hat{\bar{F}}\left(c\right)}{\hat{f}\left(c\right)},\;\;\hat{b}=\ln\hat{f}\left(c\right)+\frac{c}{\hat{k}}$$
This allows for the estimation of $B^{g}$ for the expectation of any function $g\left(\underline{x}\right)$, by replacing $B_{0},\;k$ and *b* with their estimates. In particular, for the entropy we have
(50)$${\hat{B}}_{\Phi }={\hat{B}}_{0}\left(1-\hat{b}+\frac{c}{\hat{k}}\right)$$

The data from the USA are available online thanks to the United States Census Bureau [156]; of those, we chose to use the most recent available: those for year 2019 [157], and in particular those for the entire population irrespective of particular characteristics (sex, race, etc.). The empirical probability density and tail function (probability of exceedance) estimated from the data are shown in Figure 15, and compared with the entropy maximizing exponential distribution.

Here, it is useful to remark that the detailed data cover only a small portion of the range of incomes, up to less than double the mean income. Thus, they provide little information on the distribution tail (the richest people). As a result, the extrapolation becomes very important. Without the extrapolation, the mean income (i.e., the quantity $A_{1}$, according to the above notation) is USD 30,601, and it becomes USD 53,336 after the extrapolation (i.e., after adding $B_{1}$). Note that the actual mean value, according to the source data [157], is USD 54,129, i.e., close to the extrapolated estimate, which suggests that the extrapolation model is not bad. Even more drastic is the change in the second moment: before extrapolation, it is 1.62 × 10^9^, and after it 4.74 × 10^9^, i.e., almost three times higher. The final (with extrapolation) estimate of the coefficient of variation is σ/μ=1.11, which is slightly higher than 1. The final estimate of entropy (for λ = USD 1) is Φ=11.82, and that of the standardized entropy is $\Phi_{\mu }=0.94$, which is slightly lower than 1.

Overall, the picture in Figure 15 suggests that the principle of maximum entropy with Lebesgue background measure can explain the income distribution. It is interesting that the frequency of moderately rich people, from the mean income to more than twice the mean, is somewhat overpredicted by the exponential distribution, and that of the very rich (with income more than thrice the mean) is underpredicted. The incomes of the poor and middle classes do not differ from what is predicted by the principle of maximum entropy. Remarkably, the condition $\mu_{x\geq c}/\mu =80\%$ is satisfied when $\bar{F}\left(c\right)=42\%$, close to the value 43.9% of the exponential distribution (Equation (33)) and substantially higher than 20%, thus suggesting the inappropriateness of the “80/20 rule”.

Somewhat different is the picture of Sweden, shown in Figure 16, again for the year 2019. The data, provided by Statistics Sweden [158], are more detailed than the American data, covering a range of income that reaches about nine times the mean income [159]. The estimated statistics are also shown in the figure. Here, the graph is consistent with that of Figure 14 (our toy model) up to about five times the mean income, indicating the presence of a populated middle class and suggesting an effect of the redistribution mechanisms. The standardized entropy ($\Phi_{\mu }=0.79$) and the coefficient of variation (σ/μ=0.72) are lower than in the USA, suggesting lower inequality. Strikingly, however, there is an opposite effect on the very rich, whose frequency is considerably higher than that predicted by the exponential distribution. A possible explanation involves the globalization of the financial activities of high-net-worth individuals. Again, the condition $\mu_{x\geq c}/\mu =80\%$ is satisfied for $\bar{F}\left(c\right)$ at substantially higher than 20%, namely, $\bar{F}\left(c\right)=45\%$, which is close to the value 43.9% of the exponential distribution, thus suggesting again the inappropriateness of the “80/20 rule”. Overall, the principle of maximum entropy again provides a good representation of the average behavior.

## 5. Discussion and Conclusions

We have shown that entropy is one of the most misunderstood concepts, with rich and diverse interpretations that are continuously being debated. With the transplantation of the scientific term into colloquial language, the popular imagination has loaded “entropy” with almost every negative quality in the universe, in life and in society. For example, Thesaurus.com lists as its synonyms the words breakup, collapse, decay, decline, degeneration, destruction, worsening and falling apart, while on the site wordhippo.com, the synonyms listed, in addition to those previous, amount to hundreds of words with negative meanings, including deterioration, chaos, havoc, confusion, disorder, disorganization, calamity, etc. Furthermore, in scholarly articles, there is no shortage of negative associations, as quoted in the Introduction.

There are historical reasons, as discussed in Section 2, for why the concept has generated so many negative connotations. However, at the end of the 1940s, entropy acquired a clear and universal stochastic definition that is not a related to disorder. Furthermore, at the end of the 1950s, it was complemented by the principle of maximum entropy, which lies behind the Second Law, and gives explanatory and inferential power to the concept. By now, 60 years later, one would expect that entropy should have dropped its negative meanings, and be recognized as the driving force of natural change and the mother of creativity and evolution. This has not happened. Instead, it has been used as a spectre in the social sciences, including economics and ecology, to promote neo-Malthusian ideas.

The social sciences are often contaminated by subjectivity and ideological influences, which become apparent when examined from distance, in the light of history [1,2]. Here, we explore whether maximum entropy, applied to economics and, in particular, to the distribution of a wealth-related variable, namely, annual income, can give an objective description. We show that under plausible constraints related to mean income, the principle of maximum entropy results in an exponential distribution, bounded from above if we consider an upper technological limit, but unbounded otherwise. Historically, technology has played a major role in the increase in the entropy of income. Under the current conditions, technology no longer imposes a bounding condition on the economy, but it still remains an important factor in increasing wealth.

This entropy maximizing distribution emerges when the background measure has a constant density, while if a hyperbolic background measure is used, the resulting distribution is Pareto. Based on real-world data, and in particular, those related to the world’s richest, in order to give a better image of the distribution tail, we conclude that the exponential tail is not uncommon, while the Pareto tail appears particularly in anomalous periods. Surprisingly, the latest period of the pandemic has resulted in unprecedented profits for the world’s richest, with a clear Pareto tail.

We conclude that a constant (Lebesgue) density in the background measure is natural and reasonable, and that under this measure the entropy maximizing exponential distribution is connected to a stable economy. Furthermore, we examined two different factors, both leading to a reduction in entropy and the modification of the stable exponential distribution, but in different directions. On the one hand, organized societies use mechanisms of income redistribution to minimize poverty and enhance the middle class. On the other hand, an assumption can be made that politico-economic elites try to increase their profits, thus pointing toward a Pareto distribution, which expands the poor and the very rich and reduces the middle class [3]. A thorough study of the mechanisms pushing toward both the convergence and divergence of wealth distribution has been offered by Piketty [143].

Using publicly available income data for the USA and Sweden, we showed that income distribution is consistent with the principle of maximum entropy, and in particular with exponential distribution. Yet the effect of the elites is visible, as the distribution tails exceed those of the exponential. On the other hand, the data do not support the “80/20” rule, which is consistent with the Pareto distribution (with a specific value of the tail index). Specifically, 80% of the income is not generated by 20% of the population, but by more than 40% thereof, which is fully consistent with the exponential distribution.

Overall, in this study, we have tried to dispel the “bad name” of entropy in social sciences. We emphasized its connection with the plurality of options, and we showed that increasing entropy is associated with increases in wealth. In addition, we showed that a standardized form of entropy can be used to quantify inequality.

We tried to make this paper self-contained and independent, so that even a reader unfamiliar with entropy, with only a basic knowledge of calculus and probability, could understand it. The mathematical framework we developed can be readily put to work on the simplest computational framework (e.g., a spreadsheet). The entire study is of exploratory character, and is not depending on earlier published results, as our priority was to present what we believe entropy really is, and under what conditions it could be applied to economics. Future work could open up additional options, thus increasing entropy.

## Figures and Tables

**Figure 1 entropy-23-01356-f001:**
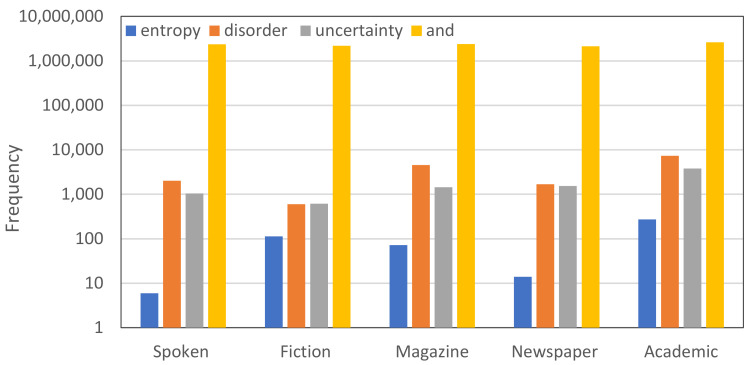
Frequency of appearances of the words entropy, disorder and uncertainty in the Corpus of Contemporary American English. The common and neutral word “and” was added as a proxy guide to the relative frequencies of the five indicated genres, which appear to be very close to each other. Data from [9,10].

**Figure 2 entropy-23-01356-f002:**
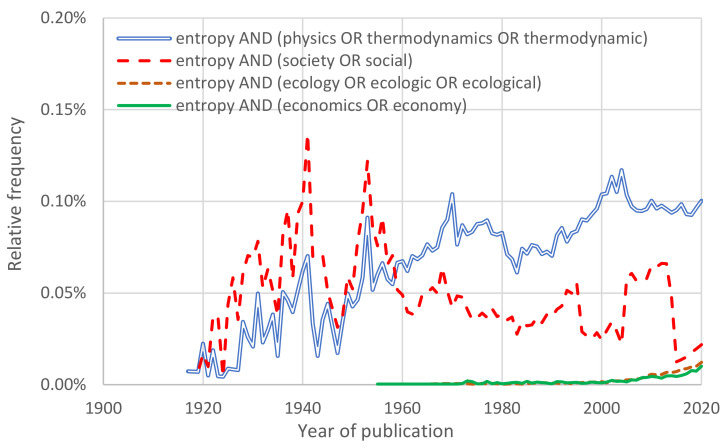
Relative frequency of appearances of the indicated key phrases in the article titles, abstracts and keywords of about 70 million articles written in English, which are contained in the Scopus database [11] up to year 2020.

**Figure 3 entropy-23-01356-f003:**
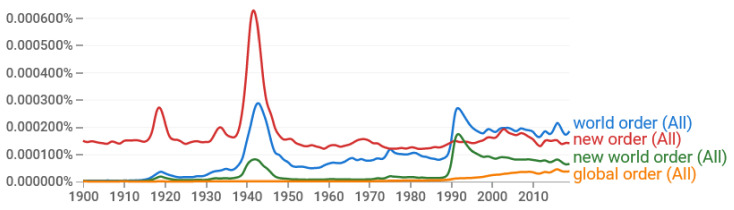
Frequency of appearance of the indicated phrases in Google Books [95,96]. Notice that the “new order” was the political order that Nazi Germany wanted to impose, and naturally its use peaked in the early 1940s. The other phrases also peaked at the same time, but they show higher peaks after 1990s.

**Figure 4 entropy-23-01356-f004:**
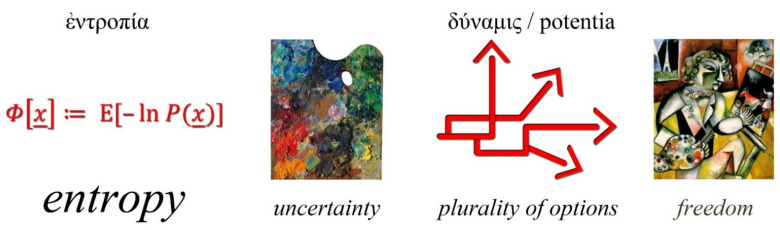
An attempt at an artistic representation of the notion of entropy. Uncertainty is depicted by Marc Chagall’s Palette (adapted from [105]) and freedom by Marc Chagall’s *Self-Portrait with Seven Fingers* [106]; δύναμις (Greek) or *potentia* (Latin) is the Aristotelian idea of potency or potentiality.

**Figure 5 entropy-23-01356-f005:**
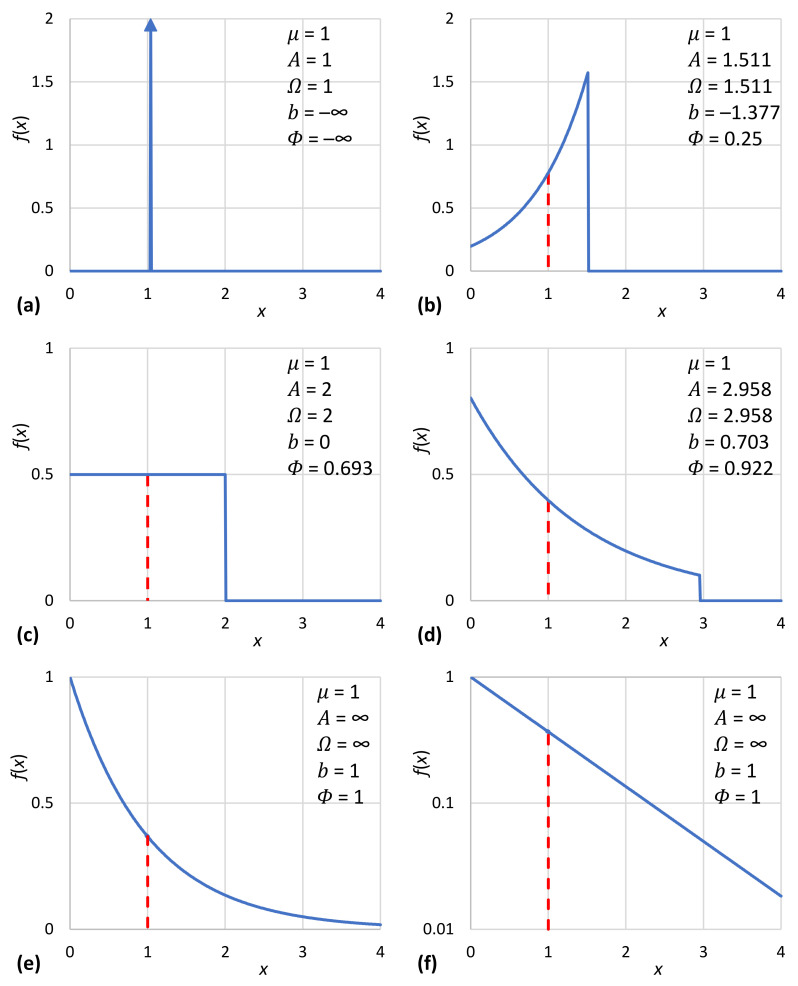
Entropy maximizing probability density functions of Table 1 for mean μ=1. Panels (**a**–**e**) correspond to cases 1 to 5 of Table 1, while panel (**f**) is same as (**e**) except that the vertical axis is logarithmic. In each panel the mean is depicted as a red dashed line.

**Figure 6 entropy-23-01356-f006:**
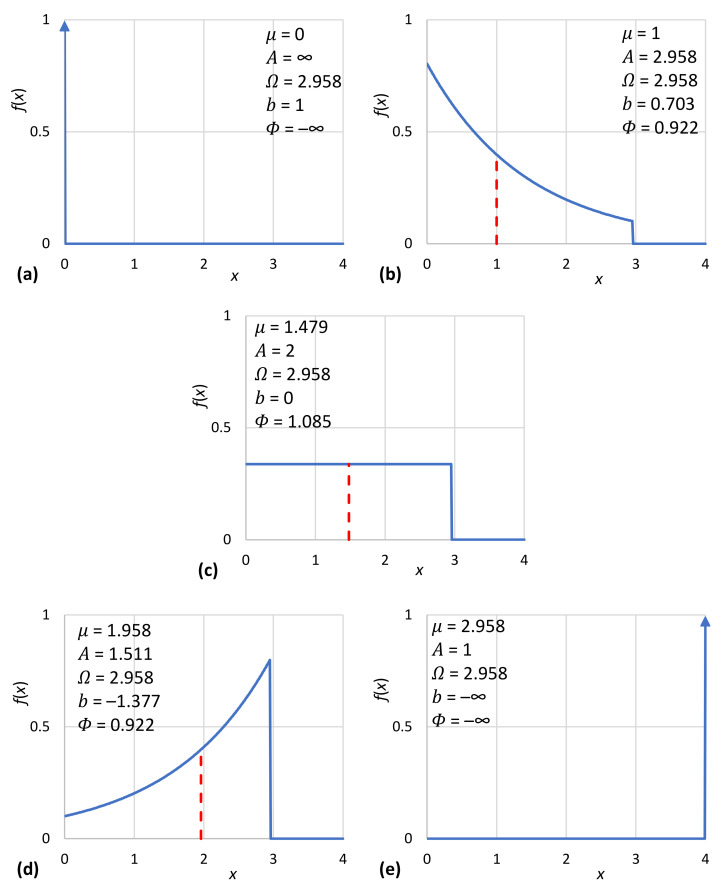
Entropy maximizing probability density functions of Table 2 for a mean technological limit Ω=2.958 (corresponding to μ=1 for case #4 in Table 2). Panels (**a**–**e**) correspond to cases 5 to 1 of Table 2 (in reverse order, so that the mean is increasing from 0 in panel (**a**) to Ω in panel (**e**)). In each panel, the mean is depicted as a red dashed line.

**Figure 7 entropy-23-01356-f007:**
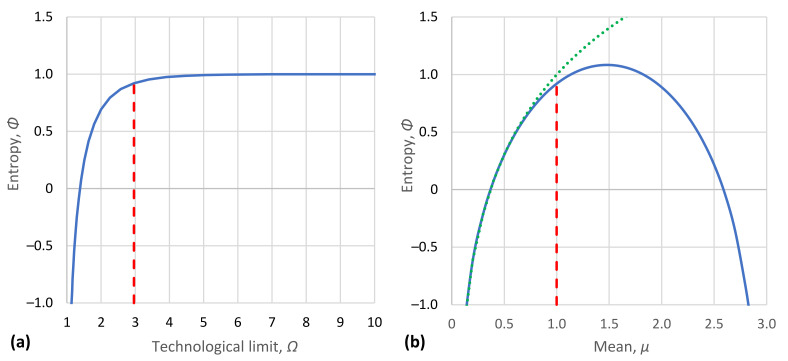
Maximum entropy as a function of (**a**) the technological limit Ω for constant mean μ=1 and (**b**) the mean *μ* for a constant technological limit Ω=2.958. The red dashed lines correspond to identical cases in the two panels. The green dotted line in panel (**b**) depicts the maximum entropy for the infinite technological limit.

**Figure 8 entropy-23-01356-f008:**
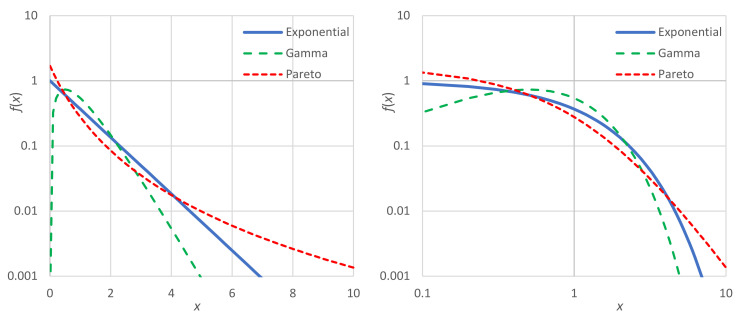
Comparison of the entropy maximizing distribution (for Lebesgue measure $\beta \left(x\right)=1/\lambda )$ with two other distributions, in terms of their probability density functions $f\left(x\right)$. The domain of x is [0,∞) and in all three distributions the mean is μ=λ=1. The entropy maximizing distribution is exponential with entropy Φ=1, and the other two are gamma with shape parameter 2 and Pareto with tail index ξ=0.408. Both the gamma and the Pareto distributions have the same entropy, Φ=0.884<1, but they have different coefficients of variation, 0.5 and 2.325, respectively. In both panels, $f\left(x\right)$ is plotted on logarithmic axis, while the variable x is plotted on linear axis in the left panel and logarithmic axis in the right panel. Notice in the left panel that the exponential density has constant slope −1, while the gamma density has slope −2 for large *x*. Likewise, in the right panel, the Pareto density has a constant slope −1−1/ξ=−3.453.

**Figure 9 entropy-23-01356-f009:**
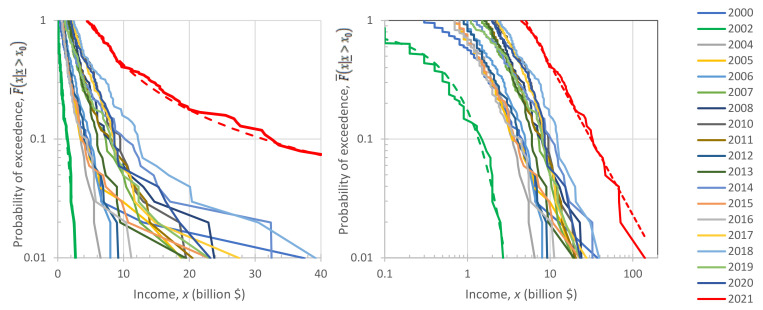
Conditional probability of exceedance of the annual income of the richest persons in the world for the indicated years. The income per person was found by subtracting the total net worth of a year from that of the previous year. For the years 2002 (lowest average income) and 2021 (highest average income) exponential and power–law trends, respectively, are also plotted with dashed lines of the same color (where in the left panel the green dashed line for 2002 is indistinguishable from the continuous line). In both panels, the probability of exceedance is plotted on logarithmic axis, while the income x is plotted on linear axis in the left panel and logarithmic axis in the right panel.

**Figure 10 entropy-23-01356-f010:**
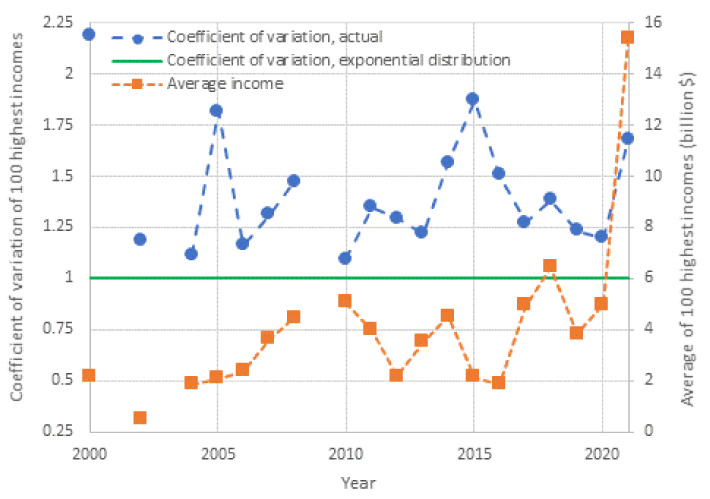
Average and coefficient of variation of the 100 highest incomes in the world per year. The coefficient of variation is calculated for the difference $\underline{x}-x_{0}$, where $x_{0}$ is the 100th highest income value, so that it can be representative for the entire population (see explanation in text).

**Figure 11 entropy-23-01356-f011:**
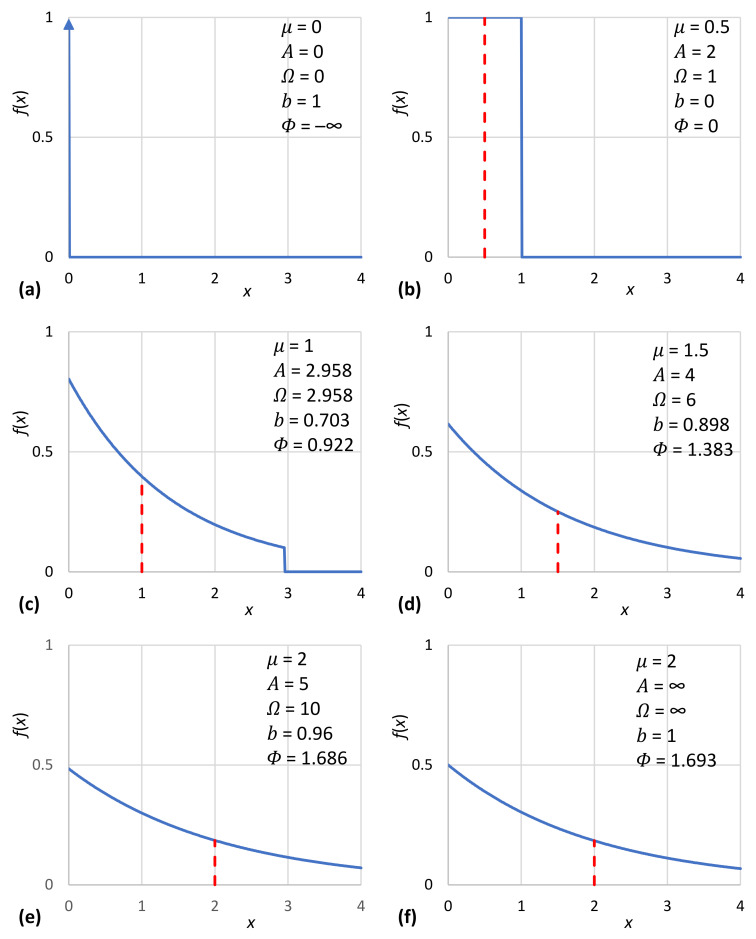
A possible evolution of human wealth from prehistory to modern times: from the primitive classless society (**a**) to a uniform distribution of wealth (**b**) and to the increasing diversity of wealth, inequality and stratification (**c**–**e**). Panel (**f**), in which the technological limit is infinite, is similar to panel (**e**), illustrating the fact that if the technological limit is large enough, its effect can be neglected.

**Figure 12 entropy-23-01356-f012:**
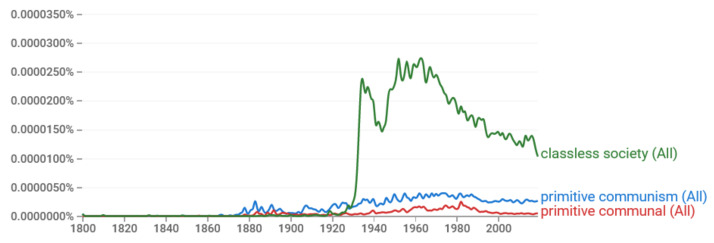
Frequency of appearances of the indicated phrases in Google Books [95,96].

**Figure 13 entropy-23-01356-f013:**
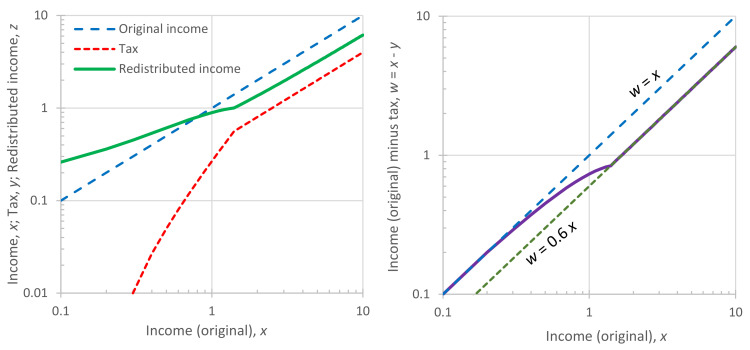
Variation in the indicated quantities with the original income (x) in the toy model: (**left**) tax and redistributed income; (**right**) original income minus tax.

**Figure 14 entropy-23-01356-f014:**
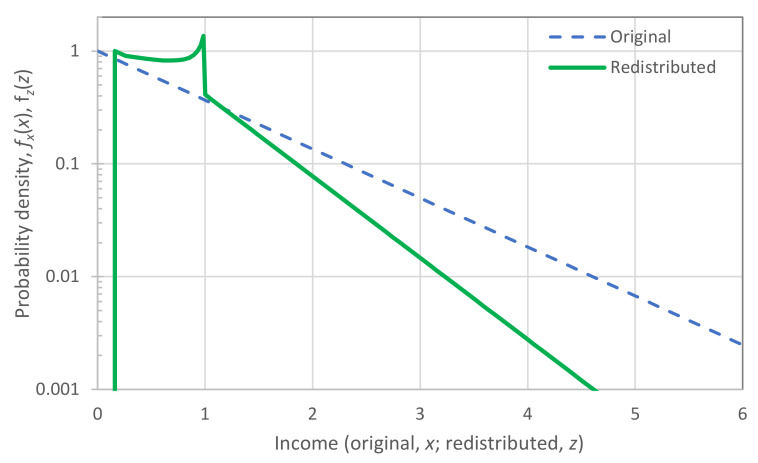
Comparison of the density function of the original income (x) and the final (redistributed) income (z) in the toy model.

**Figure 15 entropy-23-01356-f015:**
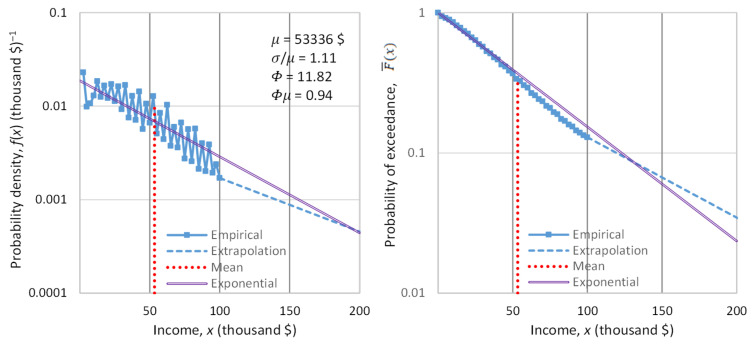
Illustration of the entropic framework using income data from the USA for year 2019; (**left**) probability density; (**right**) tail function (probability of exceedance).

**Figure 16 entropy-23-01356-f016:**
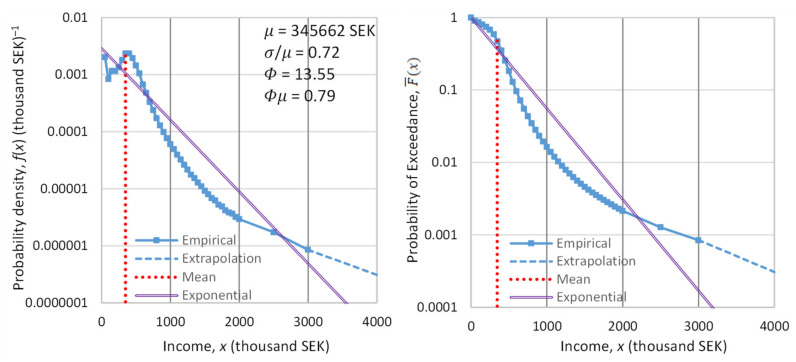
Illustration of the entropic framework using income data from Sweden for year 2019; (**left**) probability density; (**right**) tail function (probability of exceedance).

**Table 1 entropy-23-01356-t001:** Special (limiting) cases of the entropy maximizing distribution of Equation (18) for constant mean *μ* and varying technological limit Ω=Aμ, along with two exact solutions * (#2, #4) calculated by Equation (21) for $c=2^{-3}$ and $c=2^{3}$, respectively.

#	A	b	$f\left(x\right)$	$\Phi \left[\underline{x}\right]$	Distribution
1	1	−∞	$\delta \left(x-\mu \right)$	−∞	Certain (x=μ)
2	$\frac{21\ln2}{24\ln2-7}=1.511$	$1-\frac{24\ln2}{7}=-1.377$	$\frac{0.197e^{1.377x/\mu }}{\mu }$	$1+\ln\left(\frac{\mu }{\lambda }\right)-\ln\left(\frac{2^{24/7}\left(24\ln2-7\right)}{49}\right)=0.250+\ln\left(\frac{\mu }{\lambda }\right)$	Truncated anti- exponential
3	2	0	1/2μ	$\ln2+\ln\left(\frac{\mu }{\lambda }\right)$	Uniform
4	$\frac{21\ln2}{7-3\ln2}=2.958$	$1-\frac{3\ln2}{7}=0.703$	$\frac{0.803e^{-\frac{0.703x}{\mu }}}{\mu }$	$1+\ln\left(\frac{\mu }{\lambda }\right)-\ln\left(\frac{2^{24/7}\left(7-3\ln2\right)}{49}\right)=0.922+\ln\left(\frac{\mu }{\lambda }\right)$	Truncated exponential
5	→∞	1	$e^{-x/\mu }/\mu$	$1+\ln\left(\frac{\mu }{\lambda }\right)$	Unbounded exponential

* Numerals are rounded to three decimal digits.

**Table 2 entropy-23-01356-t002:** Special (limiting) cases of the entropy maximizing distribution of Equation (18) for constant technological limit Ω and varying mean μ=Ω/A, along with two exact solutions * (#2, #4) calculated by Equation (21) for $c=2^{-3}$ and $c=2^{3}$, respectively.

#	A	b	$f\left(x\right)$	$\Phi \left[\underline{x}\right]$	Distribution
1	1	−∞	$\delta \left(x-\Omega \right)$	−∞	Certain (x=Ω)
2	$\frac{21\ln2}{24\ln2-7}=1.511$	$1-\frac{24\ln2}{7}=-1.377$	$\frac{0.297e^{2.079x/\Omega }}{\Omega }$	$1+\ln\left(\frac{\Omega }{\lambda }\right)-\ln\left(\frac{3\left(2^{24/7}\right)\ln2}{7}\right)=-0.163+\ln\left(\frac{\Omega }{\lambda }\right)$	Truncated anti- exponential
3	2	0	1/Ω	$\ln\left(\frac{\Omega }{\lambda }\right)$	Uniform
4	$\frac{21\ln2}{7-3\ln2}=2.958$	$1-\frac{3\ln2}{7}=0.703$	$\frac{2.377e^{-2.079x/\Omega }}{\Omega }$	$1+\ln\left(\frac{\Omega }{\lambda }\right)-\ln\left(\frac{3\left(2^{24/7}\right)\ln2}{7}\right)=-0.163+\ln\left(\frac{\Omega }{\lambda }\right)$	Truncated exponential
5	→∞	1	$\delta \left(x\right)$	−∞	Certain (x=0)

* Numerals are rounded to three decimal digits.

## Data Availability

The databases that have been used are referred to in detail in the citations given in the text and are publicly available.

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
