# Peer review of "Entropy and Wealth"

_entropy, 2021, doi:10.3390/e23101356_

Round 1

Reviewer 1 Report

This article contributes significantly to the concept of entropy and its use in different areas of knowledge, in particular in the social sciences. It also discusses the maximum entropy principle and its application to wealth distribution. The article is well structured, provides a comprehensive list of references and arrives at interesting results. It can be a good reference for scientists who use the concept of entropy in their research. I recommend its publication.

Author Response

Please see the attachment entitled: Rebuttal to review comments on “Entropy and Wealth”

Reviewer 2 Report

The authors claim to present a stand-alone paper with some major claims, primarily, 1. denouncing the negative association of the term entropy in social sciences and colloquial usage, and 2. denouncing the need for a thermodynamic entropy. Further, they try to define entropy as a freedom of choice - which I can agree. In order to do so, they use distribution of wealth in societies and the income data to prove the concept of maximization of entropy. While I agree with some of their ideas I also disagree with some of their claims, quite strongly. These are mostly concerning their claims regarding thermodynamic entropy. I would be happy to hear from the authors about my concerns.

Fig1 – will normalizing the y-axis change something? Because it is hard to figure out what are sample sizes in each category

Line 135 - this is the first time the authors describe their point of view. I suggest moving this further up at the beginning of the introduction

The claim that the authors propose in line 137, is it general? Or, specific to the usage of the term, 'entropy'?

Line 135 – I don’t agree as in most realistic (non-equilibrium) scenarios entropy to day remains a fuzzy concept

Line 293 – clarification needed. I can understand for a simple process such as melting of ice to room temperature. Of course, the room temperature is the most probable state, so your statement makes sense. But what if I heat up a container of water? And I continue to heat it by regulating the heat flow. As the temperature rises, there will be thermal instabilities that will increase the rate of heat dissipation. So, what is the most probable state here? Is the entropy increasing? Is the most probable state known in this case? What if these instabilities give rise to complex spatio-temporal patterns? Does entropy decrease or increase, locally? globally?

Line 436 – I don’t agree

Line 438 – This is not true. Entropy can never be a fundamental quantity as it cannot be measured. It is an extensive thermodynamic property. Temperature is an intensive property that can be measured. In most situations, we have local gradients in temperature which leads to thermodynamic fluxes and forces which then lead to entropy production; and in most cases this entropy can not be analytically defined!

Line 446-456 – The possibility for entropy to spontaneously decrease exists and is not zero; yes, this possibility decreases exponentially for a large enough system size but is non-zero.

Line 1066 – Generally speaking, heavy-tailed distribution of wealth in societies among others have already been studied – rich getting richer phenomenon - for instance  

Author Response

(The authors gave the same response as above.)

Reviewer 3 Report

I had great expectation from this quite long manuscript by Professors Kousoyiannis and Sargentis, this is partly the reason I accepted it for review and this is why I did study it carefully. My final conclusions are however extremely mixed. In a short overview I would characterize this work as a combination of an incomplete attempt for a review on the concept of entropy and a disputable approach to understand social inequalities in wealth and income by using the entropy maximization principle. The manuscript in the present form is definitely not appropriate as a research article.

The very long introduction on the use of the concept of entropy in physics, information theory and social science and the personal view of the authors on this, could be the seed of a review paper IF it would be more complete and again carefully revised. In this introductory part there are many statements that are definitely disputable from a statistical thermodynamics point of view. The affirmations that the thermodynamic entropy is a totally different concept from the probabilistically defined informational entropy is somehow in contradiction with what we nowadays teach in basic statistical physics. Both the Boltzmann and Shannon formula can be elegantly derived from the thermodynamic definition of entropy using the Liouville theorem and the ergodic principle, and this is done in many elementary statistical physics textbooks. Therefore, the relation between the probabilistically defined entropy and the thermodynamically derived one are known, and some of the long philosophical discussions from the beginning of this paper is in this view meaningless. We physicist definitely agree with the authors view of entropy as uncertainty rather than disorder, and this is no new viewpoint. The basic definitions of entropy in section 2.6 are also generally known information and should not be the part of a research communication nowadays, even if one attempts a logical completeness of the discussion. Some new aspects and generalization of the concept of entropy as the Tsallis or q-entropy should be however discussed if one proposes a review on the concept of entropy in sciences. The entropy maximizing distributions are also well-known nowadays and there is way too many application of it to be mentioned here, but it should be at least properly cited in the manuscript.

The second part of the manuscript is intended to be an application of the entropy maximization principle to understand income and wealth distributions in modern societies. This could be a separate research work in case it would be revised and reconsidered in the view of recent results. When discussing this part, the authors are seemingly totally unaware of the work of many illustrious economist and the  large econo-physics literature on this subject. It seems to me quite strange to talk on this subject omitting the seminal work of Piketty (Capital in the Twenty-First Century, Harvard Univ. Press, Cambridge, Massachusetts, 2014), or the great review works of Yakovenko and  Chakraborti (Colloquium: Statistical mechanics of money, wealth, and income. Rev. Mod. Phys. 81, 1703; Econophysics of Income and Wealth Distributions, Cambridge Univ. Press, 2013, etc. ). On discussing the possible exponential trend in income distribution the very early work of Dragulescu and Yakovenko are not mentioned ( Evidence for the exponential distribution of income in the USA, Eur. Phys. J. B 20,2001, 585-589). When discussing on some modern time empirical data the authors use very poor-quality data and come to erroneous conclusions, contradicting the nowadays widely accepted Pareto-like tail of these distribution functions. The carefully analyzed exhaustive data for wealth and income are ignored and also the works that offer a simple and elegant model to understand the distribution  for the whole range of income and wealth (Z. Neda et. al, Scaling in income inequalities and its dynamical origin, 2020, Physica A, vol. 549, 124491; I. Gere et. al, Waelth distribution in modern societies: Collected data and a master equation approach, 2021, Physica A, vol. 581, 126194). The use of entropy maximization to understand social inequalities is not a new idea, it has been applied also by using the generalized q-entropy, and it led also to the right power-law tails in the distributions. As a conclusion I consider that this application part of the manuscript has to be also largely revised before any publication. The discussion of the authors on the influence of taxation and total amount of wealth and average income are definitely interesting ideas, but these has to be redone in view of what is already present in the literature nowadays. Even a simple Google search would indicate the authors many works that have done similar approaches to wealth and income inequalities.

In conclusion I would ask a major revision of this manuscript. It would be desirable to split the present work in two and publish them separately. One work would be a review and a second one some novel contributions to understand social inequalities by means of entropy maximization. For both works however the authors should consider a thorough review of the existing literature and insist much more on the scientific rather than the philosophical context. The authors should carefully separate the new results from the ones that are already known in the economics or econo-physics literature. I cannot recommend publication of the present form of the manuscript.

Author Response

(The authors gave the same response as above.)

Round 2

Reviewer 2 Report

I have read the revised manuscript. While I still don't agree with some statements in the paper regarding the idea to abandon thermodynamic entropy all together and the proposal to consider entropy as a fundamental variable along with temperature, I do feel that the paper encourages dialogue in that direction. Entropy is a highly abused term from both semantics and scientific point of view and this paper provides a platform to  atleast have a debate on that. 

Author Response

We are grateful to the Reviewer for the positive comment and recommendation.

We appreciate and share her or his attitude that possible disagreements with some of our statements would be positive, and that publication of our the paper would provide a platform to have a debate on those disagreements.

Reviewer 3 Report

I learned that the authors have taken into account some criticism from the Reviewers and made important improvements to the manuscript by coordinating their findings with the results exiting in the current literature of econophysics. In this sense  the second part of the work, concerning the application of the entropy maximization principle to understand  income and wealth distribution is scientifically much more solid. I am however still concerned with the rather long introductory part, which should be shortened. I have found out at the end of the manuscript the very sad news about Prof Xanthopuulus passing away. I just wonder whether the whole discussion and the article cannot be dedicated in his memory right from the beginning, making thus sense for this rather philosophic and long introduction. This would require just a few introductory words in the beginning, and than the reader could understand also the reason of this long and otherwise inappropiate discussion in a scientific article. In conclusion, I consider that such an approach to this longer article might be a an acceptable solution for Entropy.

Author Response

We are really grateful to the Reviewer 3 for her or his comment.  

We are very glad that she or he found our improvements to the manuscript important, and our findings coordinating with the results in the current literature of econophysics.

We embraced the Reviewer’s idea to refer to our late mentor Themistocles  Xanthopoulos from the beginning. We have made the dedication to him just below the abstract. We particularly thank her or him for this idea, which encouraged us to make more proper reference to his books and pay more appropriate honour to his memory.